# Interannual variability in the summer dissolved organic matter inventory of the North Sea: implications for the continental shelf pump

Saisiri Chaichana[1,2], Tim Jickells[2], Martin Johnson[2,3]

[1]Department of Environmental Sciences, Songkhla Rajabhat University, Songkhla, Thailand
[2]Centre for Ocean and Atmospheric Sciences, School of Environmental Sciences, University of East Anglia, Norwich, UK
[3]Centre for Environment, Fisheries and Aquaculture Science, Lowestoft, UK

*Correspondence to*: Martin Johnson (martin.johnson@uea.ac.uk)

**Abstract.** We present the distribution and C:N stoichiometry of dissolved organic matter (DOM) in the North Sea in two summers (August 2011 and August 2012), with supporting data from the intervening winter (January 2012). This data demonstrates local variability superimposed on a general pattern of decreasing DOM with increasing distance from land, suggesting concentrations of DOM are controlled on large spatial scales by mixing between the open North Atlantic and either riverine sources or high DOM productivity in near-shore coastal waters driven by riverine nutrient discharge. Given the large size and long residence time of water in the North Sea, we find concentrations are commonly modified from simple conservative mixing between two endmembers. We observe differences in dissolved organic carbon (DOC) and dissolved organic nitrogen (DON) concentrations and land-ocean gradients between the two summers, leading to an estimated 10-20 Tg difference in the DOC inventory between the two years, which is of the same order as the annual uptake of atmospheric $CO_2$ by the North Sea system, and thus significant for the carbon budget of the North Sea. This difference is not consistent with additional terrestrial loading and is more likely to be due to balancing of mixing and in-situ production and loss processes across the North Sea. Differences were particularly pronounced in the bottom layer of the seasonally stratifying northern North Sea, with higher DOC and C:N ratio in 2011 than in 2012. Using other data, we consider the extent to which these differences in the concentrations and C:N ratio of DOM could be due to changes in the biogeochemistry or physical circulation in the two years, or a combination of both. The evidence we have is consistent with a flushing event in winter 2011/12 exchanging DOM-rich, high C:N shelf waters, which may have accumulated over more than one year, with deep North Atlantic waters with lower DOC and marginally higher DON. We discuss the implications of these observations for the shelf sea carbon pump and the export of carbon rich organic matter off the shelf and hypothesise that intermittent flushing of temperate shelf systems may be a key mechanism in the maintenance of the continental shelf pump, via the accumulation ad subsequent export of carbon-rich dissolved organic matter.

**Keywords.** Dissolved Organic Matter, Carbon, Nitrogen, Stoichiometry, Mixing, Continental Shelf Pump, North Sea

# 1 Introduction

Coastal and shelf seas are generally more productive than the open ocean (Jickells, 1998; Simpson and Sharples, 2012), and through various processes may be disproportionately important for the drawdown of atmospheric carbon to the deep ocean (Bauer et al., 2013; Regnier et al., 2013; Thomas et al., 2005b; Tsunogai et al., 1999). Our understanding of the mechanisms of carbon pump processes on the shelf and their relative importance is limited by observational data, the complexities of shelf circulation and interannual variability of both biological processes and physical drivers. This is particularly the case in shelf sea systems such as the North Sea, which has complex physical circulation, involving large water exchange with surrounding seas and ocean; and strong anthropogenic forcing from surrounding land masses, especially via substantial river discharge (Bozec et al., 2005; Kühn et al., 2010; Simpson and Sharples, 2012; Thomas et al., 2004, 2005b). The formation of dissolved organic matter (DOM) from inorganic carbon during primary production can lead to uptake of $CO_2$ from the atmosphere and, depending on the lifetime and ultimate fate of the DOM, this carbon may remain out of the atmosphere many hundreds or thousands of years (e.g. Barrón and Duarte, 2015; Bauer et al., 2013).

In this study we investigate the variability of the organic carbon inventory of a large and complex shelf sea by considering the evidence provided by two high-spatial-resolution summer surveys. We consider organic matter concentrations and stoichiometry and deviations from conservative mixing between river and open ocean end members. These data were collected during cruises of opportunity during a PhD research programme, so our analysis is focussed on diagnostic, geochemical approaches to understanding bulk concentrations. It intentionally *does not* attempt to elucidate distinct sources or types of DOM, or determine process rates, given only *prima facie* evidence.

## 1.1 Dissolved organic matter and the continental shelf pump

One potentially important mechanism of shelf carbon export to the deep ocean is via DOM. In the marine environment, DOM can be a complex mixture of organic material from various sources both terrestrial and marine, with a spectrum of lifetimes from hours to millennia (Hansell, 2013; Nelson and Wear, 2014; Repeta, 2015). The reactivity of DOM and in particular its availability for breakdown by marine microbes is a key factor controlling its resistance to degradation to inorganic carbon and thus capacity for long-term carbon storage; with refractory, i.e. unreactive, DOC having a lifetime of hundreds or thousands of years (e.g. Jiao et al., 2014). A recent synthesis of DOC in continental shelf seas observes a strong inverse relationship between distance from land and DOC concentration, with significant enrichment of DOC in nearshore waters relative to the open ocean (Barrón and Duarte, 2015). This is consistent with a long term observational data set from the North Sea (Van Engeland et al., 2010) and results from the Mid-Atlantic Bight (Bauer et al., 2002; Vlahos et al., 2002) and Arctic (Wang et al., 2006). This seemingly ubiquitous gradient between low and high salinity is found by Barrón and Duarte (2015) to drive export of dissolved organic matter from shelf seas to the global ocean on a scale which is significant for the global carbon cycle.

The DOC export flux from continental shelf seas is composed of a combination of terrestrially-derived organic matter from rivers (allochthonous DOM) and autochthonous DOM produced on the shelf through in-situ autotrophic processes. The relative contributions from the different sources is uncertain, although there is strong evidence that much terrestrial organic matter is respired, photooxidised or buried in shelf sea systems. (e.g. Asmala et al., 2016; Ward et al., 2017). Shelf seas with longer water residence times will therefore tend to deliver less terrestrial (dissolved and particulate) organic matter to the open ocean (Asmala et al., 2016; Barrón and Duarte, 2015). Rivers also deliver significant nutrient loading to the coastal zone. The consequent nutrient-driven primary productivity may be a source of new DOM produced in near-coast waters, which could also lead to the relationships between DOM and proximity to land observed by the abovementioned studies.

The stoichiometry of exported DOM may also be an important indicator of the efficiency of the shelf carbon pump. Higher C:N dissolved organic material represents greater carbon fixation per unit nutrient and the export of carbon-rich organic material off-shelf in particular contributing to nutrient retention on the shelf and a more efficient shelf pump (Humphreys et al., 2018). Where autochthonous production dominates the DOM pool, the C:N ratio can be compared to the so-called Redfield ratio; and deviations of DOM stoichiometry from Redfield have commonly been observed (Abell et al., 2000; Aminot and Kérouel, 2004; Ducklow et al., 2007; Hopkinson and Vallino, 2005; Letscher and Moore, 2015; Pujo-Pay et al., 2011). However it must be recognised that particulate organic matter is not produced at a single fixed C:N ratio even in primary production (e.g. Moore et al., 2013) and so DOM which deviates from Redfield may not necessarily have been modified from the stoichiometry of the primary production from which it originates. Despite these caveats, a net enrichment of carbon relative to nitrogen as shelf seas cycle organic matter and nutrients implies re-use of the nutrient (i.e. nitrogen) to drive further primary production, thus decoupling organic matter processing from the relatively fixed stoichiometry of algal growth.

DOM that is carbon-rich relative to the Redfield ratio might be produced by the preferential remineralisation of phosphorus and nitrogen over carbon from DOM by bacteria (e.g. needing to meet their nutrient requirements for growth) (Lønborg et al., 2010) or via the 'overflow production' of carbon rich organic matter by phytoplankton under situations of nutrient limitation, also referred to as 'carbon overconsumption' (Prowe et al., 2009; Toggweiler, 1993). The bioavailability/ reactivity of exported DOM is also likely to impact the efficiency of the shelf pump. A change to more refractory (i.e. resilient to degradation) DOM would lead to accumulation in the marine system and overall net removal of carbon from the atmosphere. The bioavailability of DOM in shelf environments has been proposed to be linked to nutrient availability, with higher nutrient (specifically, nitrogen) availability leading to greater remineralisation of carbon-rich refractory DOM and thus a decrease in microbial carbon pump efficiency (Jiao et al., 2014).

The carbon to nitrogen ratio of DOM therefore has the potential to be a useful diagnostic of the state of the shelf system – indicating the efficiency of nitrogen re-use throughout the microbial food web. However, the interpretation of DOM C:N is

complicated by the interactions with river inputs at differing concentrations and probably variable stoichiometry, as well as seasonal and interannual variability. Some studies have used other parameters to elucidate sources of DOM, for example biomarkers and isotopic signatures (e.g. Kaiser and Benner, 2012) or spectroscopic or fluorescent signatures (e.g. Painter et al., 2018). In some systems the N:C ratio can be used to determine the relative contributions of terrestrial and marine sources to DOC, if endmembers are known and conservative mixing can be assumed (Perdue and Koprivnjak, 2007). In the absence of such techniques, the use of C:N alone as a diagnostic variable in shelf seas must be approached with caution. We explore in this paper its potential application to the North Sea case, in the context of other oceanographic measurements and observations.

## 1.2 The North Sea

The North Sea can be characterised by a shallow, well mixed water column in the south (Emeis et al., 2015), and a deeper seasonally stratifying system to the North (Van Haren and Howarth, 2004; Knight et al., 2002), with the boundary between these systems at roughly 55 °N and ~40m depth. The North Sea is a highly productive shelf sea, although in spite of this the southern North Sea, which remains well mixed throughout the year and is highly affected by riverine carbon fluxes may be net heterotrophic and a source of $CO_2$ to the atmosphere (Bozec et al., 2005; Prowe et al., 2009). The northern North Sea is a net sink for atmospheric $CO_2$, argued to be driven by seasonal stratification and the consequent vertical separation of surface autotrophy and respiration of exported carbon in the net-heterotrophic waters below the thermocline (Bozec et al., 2005; Clargo et al., 2015). In the 'classic' North Sea shelf pump mechanism, below-thermocline waters in the northern North Sea then exchange with the deep ocean, so the accumulation of dissolved inorganic carbon in the deeper waters of the norther North Sea and its subsequent exchange with the open Atlantic is thought to be an important mechanism in the shelf carbon pump (Bozec et al., 2005; Thomas et al., 2004).

The role of DOM in the carbon pump of the North Sea is less clear than that of dissolved inorganic carbon (DIC). Thomas et al. (2005a) observe inflows and outflows to the North Sea which suggest minor net respiration of DOC, based on observations from a single year (spring and autumn cruises). However, Prowe et al. (2009) invoke the production of carbon rich organics via overflow production in the northern North Sea to explain the apparent decoupling of nitrate and DIC drawdown over a seasonal cycle, i.e. greater drawdown of carbon per unit nitrate uptake than would be predicted by the 'Redfield' ratio of 6.6:1 C:N. Various studies also point to strong seasonality in the in-situ production of DOM by algae (especially during the spring bloom) and subsequent slow degradation (time scales of weeks to months), albeit with strong spatial variability and apparent patchiness (Van Engeland et al., 2010; Johnson et al., 2013; Suratman, 2007; Suratman et al., 2008, 2009). At the current level of knowledge and available data it is not yet possible to determine whether the North Sea is a net source or sink for DOC in a typical year. In order to better understand the dynamics and stoichiometry of DOM, and whether or not there is the potential

for an organic matter continental shelf pump for carbon (Barrón and Duarte, 2015; Thomas et al., 2005a), a more detailed understanding of the spatial variability and large-scale controls on DOM concentrations is thus required.

Unlike some enclosed coastal seas such as the Baltic or Mediterranean, the North Sea is characterised by a considerable
'through-flow' of seawater originating from, and returning to, the Atlantic Ocean. Except in the river estuaries flowing into the North Sea, and in the near-coastal zone of the southern North Sea, where river influence is greatest, salinity tends to range between 30 and 35.5. The freshest part of this range is typically found in the Norwegian Coastal Current, in surface waters to the northeast of the study area, which combines local and regional river water with brackish water from the Baltic outflow (Winther and Johannessen, 2006). Sampling the North Sea is therefore analogous to sampling the high salinity portion of a
large, extended estuary with strong ocean influence (e.g. Hydes et al., 1999), although complicated by circulation and seasonal stratification.

The horizontal gradient of salinity through the North Sea and also of any associated tracers of river or ocean, are determined by the exchange rate, or residence time of water. Water residence times in the North Sea are uncertain, but may vary between
flushing on the timescale of <1 year to almost a decade (e.g. Blaas et al., 2001; Holt et al., 2009; Otto et al., 1990) depending on prevailing physical conditions on and off the shelf. Sharples et al. (2017) find the North Sea to have one of the longest residence times of shelf seas globally, due to its large size (specifically the width of the shelf) and its relatively high latitude, meaning that Coriolis force is strong so riverine inputs take a more circuitous route to the open ocean than they do at lower latitudes. They conclude as a result of this Coriolis effect that low latitude shelf seas are likely to be much more efficient at
transporting riverine DOM to the open ocean than shelf seas at higher latitudes.

Mixing and transport of riverine material through the North Sea is ultimately driven by river flow and shelf-edge exchange of water with the open ocean. Recent studies have noted that there can be important interannual variations of North Sea circulation and exchange with open Atlantic waters linked to climate cycles such as the North Atlantic Oscillation (NAO, Salt et al., 2013;
Sheehan et al., 2017; Winther and Johannessen, 2006). This is both due to changing wind regime (strength and direction) driving shelf-edge exchange and circulation patters and also changing precipitation over land leading to greater runoff. Associated shifts in patterns of primary production  have also been suggested, which have the potential to alter the net carbon exchange fluxes through both physical and biogeochemical processes (McQuatters-Gollop et al., 2007).

Comparison of DOM concentrations with idealised (i.e. conservative with salinity) mixing lines between riverine and open-ocean end member DOC and DON concentrations has often been used to infer and even quantify in-situ loss and production rates (e.g. Asmala et al., 2016; Cauwet, 2002; Ducklow et al., 2007; Margolin et al., 2016). In combination with this, the N:C ratio is sometimes used to elucidate different sources of carbon, where DOM is known to be conservative with salinity (Perdue and Koprivnjak, 2007). Relationships with salinity can appear highly conservative, particularly in estuaries with short

residence times or in unproductive regions (e.g. Köhler et al., 2003). In other situations, they can be strongly non-conservative, typically in long residence time systems, highly productive systems or simply where only a small part of the salinity gradient is observed (e.g. Asmala et al., 2016; Cauwet, 2002). In the North Sea case, the degree to which DOM is conservative with salinity may be expected to vary considerably between periods of different circulation rate; particularly given that the system is highly productive, driven by terrestrial nutrients, and therefore high autochthonous production would be expected to substantially modify DOM concentrations in addition to the effects of terrestrial DOM degradation.

In a recent study of the North Sea in summer, Painter et al. (2018) applied fluorescence analysis to identify possible sources of DOM. In their study they identify two fluorophores (active regions of the excitation-emission spectrum), both of which decreased sharply with distance from land. One of these fluorophores was identified as representative of humic matter of terrestrial origin, which showed a strong, near linear relationship with salinity over the range 30-35. Whilst strongly associated with salinity it was not correlated with bulk DOC or DON, suggesting strong autochthonous sources dominate DOM concentrations in the open North Sea, rather than riverine sources. They attribute this to high productivity and significant marine contribution to the organic matter pools. Painter et al. (2018) conclude that very little terrestrial organic matter reaches the open ocean via the North Sea due to its size and long residence time. The second fluorophore that was generally higher nearer land was identified as being indicative of biological production of DOM. This is consistent with high levels of productivity in near-coastal seas due to riverine nutrient input and could explain the strong gradients going away from land without the need to invoke conservative mixing of terrestrial organic matter. As with the terrestrial signal, the influence of near-coast production on central and northern North Sea DOM appears to be small according to Painter et al. (2018), reinforcing the idea that in-situ production probably dominates DOM composition in these regions also.

This current study has the following aims. 1) To determine the degree to which conservative mixing can explain the observed DOM distributions in the summer and winter survey data presented and to evaluate the possible reasons for any non-conservative behaviour. 2) To establish whether the North Sea is likely to be a source of DOC to the Atlantic Ocean, whether this DOC source is lower than the global average due to the long water residence time and how variable this source might be interannually and seasonally. Finally, 3) to understand the potential role of C:N stoichiometry of the DOM on the effectiveness of carbon export from the shelf and how our results impact broader understanding of shelf pump mechanisms.

In order to provide a reference for the observations presented below, we have conducted a synthesis of DOM end member concentrations from the literature with which to compare our results (detailed in Tables S1 and S2 of supplementary material). This synthesis is summarised in Figure 1, demonstrating the DOC, DON and resulting C:N ratio of DOM with salinity assuming purely conservative mixing. Two separate ocean end-members are considered 1) 200-600m depth range in adjacent regions of the Atlantic and 2) those of deeper (>600m) waters in the same region. The shallower depth range is assumed to be representative of winter concentrations in the surface Atlantic, relatively unaffected by in-situ seasonal production of DOM.

Deeper values are likely to be more representative of older waters and the marine 'background' DOM (e.g. Hopkinson and Vallino 2005). Depending on the prevailing physical conditions and season, water moving from the open ocean onto the shelf may be of shallower or deeper origin (e.g. Blaas et al., 2001; Holt et al., 2012). The calculated C:N ratios presented in Figure 1 have a similar range to those directly observed by studies used in the synthesis, which typically sit in the rage 11 to 16 (Aminot and Kérouel, 2004; Table S2 in supplementary material).

## 2 Study area, sampling and analytical methods

### 2.1 Study sites and field sampling processes

Sampling (Figure 2) was conducted on three cruises of opportunity on board the RV Cefas Endeavour in August to September 2011, January 2012 and August 2012 (Table 1). The two summer cruises were part of the long-term International Council for the Exploration of the Sea (ICES) bottom trawl survey of fisheries (ICES, 2012). The sampling grid was determined by the needs of this survey, and aimed to occupy one station in each of the ICES survey rectangles, with each station being close to the same location each year (ICES, 2012). Due to time and weather constraints the more northerly stations of the survey (North of approximately 58 °N) were not sampled in summer 2012 (Figure 2). The winter cruise surveyed a set predominantly of coastal sites in the well mixed southern and western North Sea.

The nature of the survey meant that only two water samples were taken at each station. Surface and bottom waters were sampled from 10 litre Niskin bottles attached to a CTD rosette. In general, surface and bottom water samples were collected at 2-4 meters below the surface and 5-6 meters above the seabed respectively. The cruise track focused on the open waters of the North Sea (minimum observed salinity at DOM sampling points =30.9, 87% of samples had salinity >34.0) and did not sample near shore regions with stronger riverine influence. Samples were collected at all stations for measurements of DOC and DON, along with dissolved inorganic nutrients (nitrate+nitrite, phosphate, ammonium and silicate). Additionally, chlorophyll $a$, particulate organic carbon (POC) and particulate organic nitrogen (PON) samples were collected during the summer 2012 cruises. Standard hydrographic measurements including temperature and salinity were recorded by the ship's water column profiling equipment and processed using standard techniques.

### 2.2 Analytical procedures

Water samples were filtered on-board upon collection to separate dissolved material (DOC, DON and inorganic nutrients) from particulate material (POC and PON). Filters and most of the filtration unit were combusted in a muffle furnace before use (Kaplan, 1992; Sharp et al., 1993). Gentle vacuum filtration (~5 kPa) was used through pre-combusted (450 °C, 5 hours) 47 mm diameter glass fibre filters of nominal pore size 0.7 μm (GF/F) with an ashed (550 °C, 5 hours) glass filtration unit. Filtrates were then collected in polypropylene sample tubes (Fisherbrand[TM] polypropylene centrifuge tube 50 mL, Fisher

Scientific, UK) for DOC, total dissolved nitrogen (TDN) and inorganic nutrients analysis. These filters and tubes have previously been shown to preserve these analytes without contamination (Chaichana, 2017; Suratman, 2007; Tupas et al., 1994). Filters were wrapped in aluminium foils for POC and PON analysis. Chlorophyll *a* samples were collected on the same type of GF/F glass fibre filters (without combustion, gentle vacuum filtration ~10 kPa). All samples (seawater and filters) were immediately frozen at -20 °C after filtration on board (-60 °C for chlorophyll *a* samples) until further analysis in the laboratory.

Measurements of dissolved inorganic nutrients (nitrate, ammonium, phosphate and silicate) were performed using standard spectrophotometric methods (Kirkwood, 1996) using a Skalar San$^{++}$ autoanalyser (segmented flow analysis and colourimetric chemistry, Skalar analytical, Netherlands). The limit of detection for nitrate, ammonium, phosphate and silicate were 0.1, 0.2, 0.1 and 0.1 µM, respectively. Accuracy was assured by the use of Environment Canada certified reference materials with average measured values all within 10% or better of the expected value. Note that nitrate was measured as total nitrate plus nitrite, and nitrite assumed a small fraction, as is common for oxic waters (e.g. Hydes et al., 2001; Painter et al., 2018; Suratman et al., 2008).

DOC and TDN were determined by the high temperature catalytic oxidation method coupled with nitrogen chemiluminescence detector system (high temperature catalytic oxidation (HTCO) - total organic carbon (TOC) - nitrogen detector (ND) system or HTCO-TOC-ND system) using a Skalar Formacs$^{HT}$ combustion TOC/total nitrogen (TN) analyser (Skalar analytical, Netherlands) coupled with a Skalar nitrogen chemiluminescence detector (ND20, Skalar analytical, Netherlands). The methods are similar to those we have reported previously (Johnson et al., 2013; Suratman et al., 2010) and are repeated here in brief. Samples were oxidised over a catalyst at a combustion temperature of 750 °C. The catalyst was a layer of cobalt-chromium (CoCr, ~15 g sitting on quartz wool to prevent loss of the catalyst from the bottom of the combustion tube) and cerium oxide (CeO$_2$, ~2.5 g on top) in a quartz glass column. Carbon-free, high purity air (Zero grade air, BOC gases, England) was used as carrier gas (240 mL/min). To eliminate inorganic carbon in samples, the TOC/TN analyser was programmed to add 100 µl of 10% hydrochloric acid to each 6 mL sample, sparge using pure air for 240 s and stir for 180 s. The sample injection volume was 200 µL and the best two injections were chosen automatically from up to four injections to achieve the coefficient of variation (CV) better than 2 %.

Calibrations of the instrument were carried out by potassium hydrogen phthalate (KHP) (0 – 300 µM) and a mixture of ammonium sulphate and potassium nitrate (0 – 50 µM) dissolved in Milli-Q water for DOC and TDN analysis respectively. The system blank was estimated by using acidified and sparged Milli-Q blank injection which was 29.2 ± 4.2 µM (n = 78) for DOC and was 0.6 ± 0.6 µM, (n = 79) for TDN analysis. The value agrees with the system blank reported in other previous studies (Álvarez-Salgado and Miller, 1998; Badr et al., 2003; Benner and Strom, 1993; Hopkinson et al., 1993; Koike and Tupas, 1993; Suzuki et al., 1992). The blank correction was applied to DOC and TDN data. Precision is expressed here as a CV obtained by replicate measurement of the same sample (Miller and Miller, 2010). A continuous run of 40 standards in the

same batch was used to determine analytical precisions of about 2% and 4% for DOC (used 100 μM) and TDN (used 10 μM standard) analysis, respectively.

Two types of consensus reference materials were used to verify the DOC and TDN measurements: low carbon water (LCW) and deep seawater reference water (DSR), provided by the Hansell laboratory, the University of Miami (Hansell, 2005). The consensus concentration of DOC in LCW is 1 μM. The analysis of LCW in this study yielded a small negative value of DOC concentration as the DOC value for LCW was lower than the system blank value, therefore, sample concentrations were not corrected by the LCW value. The analysis of the DSR yielded mean concentrations of $42.6 \pm 2.9$ μM (n = 98) and $42.4 \pm 2.6$ μM (n = 66), which were in good agreement with the consensus values of $41 - 44$ μM (batch 10 lot# 05-10, batch 13 lot# 02-13) and $42 - 45$ μM (batch 14 lot# 01-14), respectively. The quantitative recovery of DSR relative to the consensus values was ($100 \pm 7\%$, n = 98) and ($98 \pm 6\%$, n = 66). The oxidation efficiency of KHP relative to urea was 93-105% ($100 \pm 3\%$, n=50) and $97 - 108\%$ ($100 \pm 2$, n=20) for 50 μM and 200 μM, respectively. This agrees with previous findings which showed the result ~100% for 200 μM KHP-glycine standard compared to urea (Watanabe et al., 2007). The limit of detection (LOD) estimated as the analyte concentration giving a signal equal to the blank signal plus three standard deviations of the blank (Miller and Miller, 2010) was 6 μM for DOC analysis. The mean of DOC concentrations for DSR was similar between the analyses for the two years, with average values differing by less than 1 μM and does not show a significant difference (t-test, $p > 0.05$) between the analysis for summer 2011 and summer 2012 (Figure S1, supplementary material).

For TDN, the analysis of LCW yielded a concentration of 0 μM (no peaks were detected during TDN analysis), which was in good agreement with the consensus values of 0 μM. The analysis of the DSR yielded a mean concentration of $32.8 \pm 1.7$ μM (n = 176), which was similar to the consensus value of $31 - 33$ μM. The quantitative recovery of DSR relative to the consensus values is $102 \pm 5\%$ (n = 176). The oxidation efficiency of the mixed standard relative to urea was $92 - 106$ ($98 \pm 4$, n = 52) and $94 - 105$ ($99 \pm 3$, n = 32) for 10 μM and 50 μM, respectively. The limit of detection for TDN analysis was 1 μM. DON was calculated by subtracting dissolved inorganic nitrogen (DIN) concentration (the sum of nitrogen in the form of nitrate and ammonium) from the TDN concentration. Therefore, the precision of DON is affected by the sum of uncertainties in TDN and DIN analysis. In this study, the precision of TDN, nitrate and ammonium, calculated as a coefficient of variation obtained by replicate measurements of the same samples (Miller and Miller, 2010) were 4%, 3% and 4%, respectively. The resulting uncertainty on DON depends on the relative concentrations of these analytes in a given sample (Saunders et al., 2017), but where DIN is low (e.g. in surface waters in summer) it roughly equates to the uncertainty on the TDN analysis (4%), whereas where DIN is high and DON is relatively low (e.g. deeper stratified North Sea water and all winter samples), uncertainty can be significantly higher, up to +/- 30% at very low DON concentrations (less than 5 μM). However, in the context of the broad range of concentrations observed in this study, such uncertainties do not affect the findings or conclusions presented here.

For POC and PON analysis, filters were placed overnight (12 hours) in a desiccator saturated with concentrated hydrochloric acid (HCl, 36% w/v) fumes to remove inorganic carbon (carbonate). The filters were then dried for 24 hours at 60 °C. POC and PON were determined by the dry oxidation method using the Exeter Analytical CE440 Elemental analyser (Exeter analytical Ltd.,UK). The precision as the CV was about 1% for C and 4% for N. Chlorophyll *a* filters were extracted by acetone and measured by spectrofluorometric method (Holm-Hansen et al., 1965; Parsons et al., 1985) using spectrofluorometer (Perkin Elmer LS45, USA). The detection limit of the chlorophyll *a* measurement was 0.1 μg/L.

## 3 Results

### 3.1 Physical oceanographic conditions

During the summer cruises there was a clear difference between southern, well mixed waters and northern, stratified waters in both physical and biogeochemical parameters, as expected. By comparing surface and bottom temperatures we divide the summer cruise data up into three sets (henceforth 'water types') for further analysis: 'southern well-mixed', 'northern (stratified) surface layer' and 'northern (stratified) bottom layer'. This approach to water mass classification is commonly used in synoptic scale studies of the North Sea (e.g. Queste et al., 2013). We consider the limited winter cruise data as a whole, on the basis that it is mostly in the permanently well mixed southern part of the North Sea and given that the whole system is vertically well-mixed in winter.

Surface waters were generally warmer in summer 2012 than in 2011 (Table 2; Figure 3), and the stratified bottom waters show mostly colder but more variable temperatures in 2011 than in 2012. Temperature-salinity (T-S) diagrams (Figure 3) of the whole dataset reveal a dominant pattern of mixing of warmer, fresher water with colder, saltier water in summer, with some colder, fresher water observable in the winter data, highlighting the disproportional seasonal warming / cooling on the shelf relative to the open ocean (Tsunogai et al., 1999). Summer salinity distributions in both years reveal a strong east-west gradient in surface water salinity with the lowest salinity waters (of 31-33) being found along the east coast of the North Sea and particularly in the region of the Nowegian Coastal Current (Figure 5d). In bottom water samples, the pattern was of higher overall salinity and a south-east to north-west salinity gradient.

### 3.2 Inorganic nutrients

Inorganic nutrient concentrations, summarised in Table 2 and Figure 4, show the typical pattern expected for the North Sea, with relatively high concentrations in the winter, and at depth in the northern North Sea, and low concentrations in summer surface waters due to utilization by phytoplankton. Northern surface waters tend to be more depleted in nutrients than southern well mixed waters, due to the greater re-supply of nutrients from the benthos, river and possibly atmospheric inputs to the latter. Figure 4 presents nitrate, phosphate, ammonium concentrations by cruise and water type. Nitrate and phosphate

concentrations and inorganic N:P ratios in the northern bottom waters were significantly higher in 2011 than 2012. This suggests differences between the two years which may reflect differences in water exchange with the open Atlantic, differences in productivity, plankton community and nutrient limitation in the spring and summer before sampling or a combination of all of these. We investigated the potential bias due to the more northerly extent of the 2011 cruise potentially sampling waters richer in nitrate and phosphate. However statistical comparison (by t-test, $p < 0.05$) of only the region of the 2011 cruise south of 59°N (i.e. the section covered in both years), reveals a similar, and still statistically significant, difference (t-test, $p < 0.05$). Similar statistical comparisons of only the region south of 59°N have been made for DOC and DON; where we report differences between years these are significant even after comparison on the same latitudinal basis.

Ammonium concentrations were high (median values >1 µM) and variable in northern deep waters and not significantly different in the two summers, probably suggesting high levels of remineralisation and heterotrophy in deep waters in both years. In well-mixed waters in 2012 most observations were at or below detection limit. However, in summer 2011 concentrations in well-mixed waters were substantially elevated with a median concentration comparable to that of northern bottom waters (~1.2 µM). Ammonium concentrations have previously been suggested as being diagnostic of the trophic state of the marine system, with periods of decoupling of respiration from photosynthesis leading to accumulation in ammonium concentrations over large areas of the ocean (e.g. Johnson et al., 2007). Higher concentrations were generally observed in southern well-mixed waters in 2011 at the bottom of the water column and not in the surface. This is possibly indicative of a strongly net heterotrophic state, either due to remineralisation of sinking organic material, or resuspended benthic material in the southern North Sea in 2011 that was not the case in 2012. These elevated ammonium concentrations coincide with phosphate concentrations being higher in well mixed waters (surface and bottom) in 2011 than 2012. This elevated phosphate is also consistent with greater heterotrophy relative to autotrophy in 2011 than 2012, or potentially the influence of benthic resuspension.

**3.3 Dissolved organic carbon and nitrogen concentrations**

Figure 5 (a-c) shows the geographic distribution of dissolved organic carbon and nitrogen over the survey area in both summers. Statistical summary of the concentrations is presented in Table 3. In general, higher concentrations were seen in the coastal zones particularly in the south and east of the study area, representing the regions most influenced by river and possibly Baltic Sea inflows. Lower concentrations were generally observed in higher salinity waters in the North. This is consistent with previous observations in the Southern North Sea (Van Engeland et al., 2010) and the summer North Sea distribution pattern reported by Painter et al. (2018); and implies that mixing of riverine/estuarine DOM with open ocean waters is a major control on DOM concentration across the gradient from river to open ocean. The lowest salinity waters observed are found around the Baltic inflow to the North Sea (Figure 5d) in surface waters and are asssociated with intermediate DOC concentrations.

Figure 6 presents DOC and DON concentrations with salinity and further supports the idea that the gradient in DOM from low salinity, river influenced water to high salinity open-ocean water is a major characteristic of DOM concentrations in the North Sea. Superimposed on the simple mixing relationship is considerable variabilty in DOM concentrations. The inverse relationship between salinity and DOC is particularly clear in southern well mixed waters and northern bottom waters, with surface waters in the stratified region showing considerable deviation from the trend, particularly in the low salinity waters in the Norwegian Coastal Current (NCC). All data in our study with S<33.5 occurs in the NCC, which appears to have a much shallower DOC/salinity gradient than the other waters. Similar patterns are observed for DON in summer 2011 but in summer 2012 (Figure 6d) the salinity relationship is more confused, with northern stratified bottom waters appearing to show a positive relationship with salinity (albeit over a very limited salinity range). The near-coast winter data (Table 3) show high, but also highly variable DOC concentrations. DON concentration in winter was of a similar magnitude to the summer values but was also highly variable.

DOC average concentrations are an order of magnitude smaller than DIC (e.g. Clargo et al., 2015), and POC average concentrations are approximately six times smaller than DOC (Table 3). Hence DIC dominates the carbon inventory in the water. DON, however, becomes the dominant form of nitrogen in summer and hence its degradation rate has the potential to influence nutrient availability and phytoplankton productivity. PON is about three times smaller than DON in terms of average concentrations, further demonstrating the important role of DON as a nitrogen reservoir in nutrient cycling in shelf environments.

By regression of the DOM vs salinity relationships, it is possible to derive apparent zero salinity end-members and associated uncertainties for DOC and DON and these vary by water type and season (Table 4). The estimated zero-salinity concentrations were 420 $\mu$M DOC and 40 $\mu$M DON for the two summer surveys combined, giving a source C:N ratio of about 10. These values are consistent with either river-derived (allochthonous) DOM, or DOM produced in estuaries or the near-coastal zone (autochthonous), fed by the riverine delivery of inorganic nutrients. Given the low range of salinities observed, the high in-situ variability and the seemingly different relationship with salinity in the NCC, the end member values derived here are subject to large uncertainty, but nonetheless their values are consistent with typical global riverine DOC and DON observations (Agedah et al., 2009; Markager et al., 2011; Neal and Robson, 2000) and also with the synthesis of North Sea river end members in Table S1 of the supplementary material. The slope (~10) and y intercept (~420) for the DOC vs salinity relationship estimated here for the North Sea are also quite similar to those estimated by Bauer et al., 2013 (11.5 slope 455 intercept) for the Mid-Atlantic Bight region off the US east coast.

DOC and DON are generally well correlated with each other in summer data (Figure 7) with R-squared values of 0.62 and 0.27 for 2011 and 2012 respectively (both relationships significant at p <0.001). Mean C:N ratios of DOM (approximately 9 to 17, Table 3) were comparable to those previously reported in in the southern North Sea (10.8 – 14.8) (Van Engeland et al.,

2010) and other continental shelf waters (11 – 19) (Bates and Hansell, 1999; Hansell et al., 1993; Hopkinson et al., 1997, 2002; Kim and Kim, 2013; Wetz et al., 2008). The weaker relationship in 2012 appears to be due to high salinity waters having lower DOC and higher DON concentrations than waters of similar salinity in 2011 and is thus associated to the apparent increasing DON with salinity observed in northern bottom waters in Figure 6.

### 3.4 Interannual differences

Consistent with the strong spatial coastal to open ocean gradients, higher DOC concentrations are observed in southern well mixed than northern stratified waters in both 2011 and 2012, although this is not the case in 2012 for DON, where average concentrations in all waters of survey area are very similar (Figure 7, Table 3). In 2011 southern well mixed waters

demonstrated statistically significantly (t-test, $p<0.05$) higher concentrations of DOC and DON than either northern surface or bottom waters (Table 3, Figure 7). In the northern stratified waters DOC and DON concentrations were generally slightly higher in surface waters than at depth (statistically significant only in 2012). Larger differences in DOM concentrations however, are observed between the two years than between surface and bottom concentrations within either of the years.

The survey in 2011 presented significantly higher concentrations of DOC and DON than the summer 2012 survey in the whole water data set (all stations) (t-test, $P < 0.05$), each data set separated by the three different water types (t-test, $P < 0.05$), and each data set separated by two water types (whole surface data and whole bottom data) (t-test, $P < 0.05$). Figure 6 demonstrates that these differences are not related to sampling different salinity ranges on the same DOM/salinity gradient, but rather that concentrations of DOC for all water types, and DON for all but northern bottom waters are elevated in 2011 vs 2012 for a

given salinity. Given the challenges associated with DOM analysis it is essential that we demonstrate that this apparent difference cannot be due to a systematic analytical error in DOC measurement. We have demonstrated (Section 2, Figure S1) that repeat measurements of consensus reference materials do not vary systematically in their DOC concentration between the analyses for 2011 and 2012 samples, so we are satisfied that this is not the case and we conclude that this DOC enrichment is a real feature of the North Sea in 2011.

### 3.5 DOM in bottom waters

We investigate the change in bottom water DOM concentrations with water column depth in Figure 8, which reveals differences between the years. DOC concentrations are elevated throughout bottom waters in 2011 relative to 2012 (typically 20 to 40 µM higher concentrations than in 2012). DON in bottom waters in 2012 shows little variation with depth of water

column. However, DON in 2011 behaves very differently, with shallower waters showing enrichment in DON relative to the following year (typically 10 µM compared to 5 µM in 2012), but with deep waters showing similar or even lower DON

concentrations than in 2012. Consequently, deep bottom waters with relatively high DOC and low DON in 2011 show elevated C:N ratios (Figure 8c); in a substantial number of cases higher than the typical North Atlantic end member of ~13-15.

## 4. Discussion

### 4.1 Is DOM conservative with salinity?

5  The data presented above demonstrates that mixing between high DOM, lower salinity coastal waters and low DOM, higher salinity ocean waters is an important component of DOM dynamics in the North Sea. This does not necessarily mean that DOM is conservative with salinity, however. Figure 9 a) and c) compare our observations of DOC and DON respectively with the range of values consistent with conservative mixing based on our end member synthesis (Figure 1). We can consider the data in two distinct subsets based on salinity range: all salinities of less than 33.5 observed during this study were associated

10  with surface waters in the NCC, which is strongly influenced by Norwegian rivers and Baltic outflow (Winther and Johannessen, 2006). Salinities greater than 33.5 are all outside the NCC and can be considered to synthesise the high salinity portion of the river-ocean continuum across the wider North Sea.

The DOC and DON concentrations in the NCC do not appear to sit on the same mixing line as the rest of the corresponding

15  year's data, with concentrations well below what would be expected based on the gradients in the high salinity data. This may be due to lower DOM concentrations in the river inputs from the Norwegian rivers, or potentially photochemical degradation in the shallow, salinity stratified surface waters of the NCC. Alternatively, this may represent the influence of surface brackish water outflow from the Baltic.  We do not have the data available to resolve this, so for the rest of the discussion we focus on the waters of the main part of the North Sea with salinity of 33.5 and higher.

As noted earlier, the DOC-salinity relationship seen here is similar to that seen in the Mid-Atlantic Bight (e.g. Vlahos et al., 2002), which drains a catchment with some climatological and land use similarities to the North Sea. Vlahos et al. (2002) find an apparent zero salinity endmember of ~400 µM DOC, and an open Atlantic endmember of 47 µM, so conservative mixing lines predicted in both their study and this one are very similar (Figure 9b). Furthermore, they observe a typical excess of DOC

25  above the conservative mixing line of 10 to 40 µM of DOC. This leads to observed DOC concentrations at high salinity (>35) of greater than 50 but less than 100 µM, consistent with our summer 2011 data, but somewhat higher than the 2012 summer observations presented here. Overall there is good agreement between Vlahos et al. (2002) data and our 2011 data across a similar range observed salinities (Figure 9b).

30  By averaging observations across different salinity 'bins' - an approach previously taken by Barrón and Duarte (2015) - we can see trends in the data which are otherwise obscured by the high variability. Mean DOC with salinity (Figure 9a) shows

distinctly different trends between years: in 2011 mean values lie slightly above the highest predicted conservative mixing line but lie broadly parallel to it, whereas summer 2012 data is systematically lower and strongly consistent with the range of conservative mixing values predicted for mixing to a deep Atlantic endmember. Mean DON (Figure 9c) is broadly consistent with predicted range of conservative mixing vales, however the uncertainty on the open ocean endmember values are large compared to DOC so it is difficult to draw any conclusions from this. Nonetheless there is a clear difference between summer 2011, where minimum DON is associated with maximum salinity and summer 2012 where minimum DON concentration occurs a salinity of around 34.5 and then increases to a higher value at open ocean salinity. Furthermore, the summer 2012 pattern is similar to the data from the intervening winter, although in winter the overall concentrations are higher. A minimum in DON concentration at salinity of ~34.5 suggests a possible sink for DON, associated with an open ocean source of DON to the shelf waters, in 2012. Overall, the trends observed in summer 2012 in DOC, DON and DOM C:N with salinity, above S=34.5, suggest mixing with deep Atlantic water containing the most N-rich/ low C:N DOM that could possibly be consistent with the data collated in our open ocean end member synthesis (Table S2).

Observations of DOC and DON tend to be *consistent with or above* predicted conservative mixing values in 2011, but *consistent with or below* predicted conservative mixing values in 2012 (Figure 9). This is potentially indicative of greater net autochthonous production of DOM in the waters sampled in 2011 and greater net consumption of DOM in 2012. Both of these processes are expected to occur in the North Sea. Painter et al. (2018) see strong indication of breakdown of terrestrial and coastally-produced DOM in the wider North Sea; and multiple studies observe or predict in-situ production and degradation of DOM by plankton and bacteria in the North Sea during spring and summer (Van Engeland et al., 2010; Johnson et al., 2013; Suratman, 2007; Suratman et al., 2008, 2009). In this, as in other datasets (e.g. Johnson et al., 2013; Painter et al., 2018), no direct relationship between either chlorophyll *a* or POC/N (data not shown) was evident, demonstrating the complexity of the multiple sources, sinks and lifetimes of DOM.

Regressions of DOC vs DON for both summers (Figure 10) reveal a gradient of between 6.5 and 7. The concentrations of DOC and DON at zero salinity (from river data synthesis in Table S1 and apparent freshwater endmembers from regression analysis in Table 4) are more than an order of magnitude higher than the open ocean concentrations, so the slope of this line is, to first order, representative of the stoichiometry of DOM at low salinity. Thus the gradient could be indicative of autochthonous production of DOM by phytoplankton at or near the Redfield C:N of 6.6:1, in estuarine or coastal waters, driven by riverine nutrient input. However, it could also be consistent with freshwater DOM inputs: Mattsson et al. (2009) note in a synthesis of riverine DOM C:N ratios that rivers flowing through catchments with high levels of intensive agriculture such as the southern North Sea commonly have C:N of <10, likely due to the high inputs of inorganic nitrogen in fertiliser enriching DOM with nitrogen. Thus the C:N of ~7 is potentially consistent with an allochthonous (i.e. terrestrial) DOM source also.

We conclude from the above analysis that DOM is not conservative with salinity in the North Sea i.e. simple dilution of riverine DOM with open ocean water cannot be assumed. However, much of the deviation from conservative mixing, due to production and degradation processes is evened out by averaging to reveal trends with salinity that are broadly consistent with what would be expected from conservative mixing. Given that we expect most riverine DOM to be degrades relatively close to the coast (e.g. Painter et al. 2018) we suggest these trends are likely related to autochthonous production of DOM driven by nutrient availability, which follow a similar decreasing trend with salinity (e.g. Hydes et al. 1999). The interannual differences are substantial, with summer 2012 data indicating both a different open ocean endmember value (more consistent with deeper waters of the North Atlantic) and less autochthonous production of DOM compared to 2011.

## 4.2 Does the North Sea export organic carbon to the open ocean?

Conservative mixing of freshwater DOM (or any other tracer) with the open ocean, by definition, must result in a concentration equal to that of the ocean end member at open ocean salinity (i.e. $\geq 35.5$). Direct exchange of water of equal salinity between open ocean and shelf waters under this situation would not result in any net DOM transport off-shelf. However there must be net transport of DOM from fresh to ocean waters in conservative mixing as there is a concentration gradient along which mixing occurs. The rate of any such conservative-mixing driven export must be directly related to the rate of mixing (i.e. water exchange) through the system in question. Any elevation of DOM concentrations on the shelf at open-ocean salinity (i.e. $S \geq 35.5$) must represent an autochthonous on-shelf source that will lead to net 'enhanced' export of organic matter to the open ocean if this water were to be exchanged across the shelf break. Figure 9a demonstrates that summer 2011 satisfies this enhanced export: any mixing of water at the shelf break will result in export of DOC. Given that summer 2012 appears to be quasi-conservative above S=34 for DOC (i.e. binned average data in Figure 9a sit in a straight line very close to the predicted conservative mixing line), this implies that the export of DOC will be lower for given mixing rate than for 2011. As the DON-salinity gradient is positive above S=34.5, this suggests a net on-shelf transport of nitrogen in DON at this time. However, the exchange across the shelf break in summer tends to be low compared to winter so summer fluxes might be expected to be small irrespective of the magnitude or direction of the concentration gradient. In this case it might be more appropriate to think of the 2012 case as 'recently mixed' with the open ocean vs the 2011 case as representative of a period of time when autochthonous production was rapid relative to mixing. The low DON at salinity of around 34.5 would then be representative of in-situ degradation during the summer period, presumably of relatively bio-available, N-rich DOM of marine origin.

The global synthesis of shelf sea DOC concentrations by Barrón and Duarte (2015) leads to much greater predicted enhancement of shelf DOC concentration at $S \geq 35.5$ relative to open ocean concentrations than those observed in summer 2011 in the North Sea, or in the Mid-Atlantic Bight (MAB) in the data of Vlahos et al. (2002), as demonstrated in Figure 9b. Given the relatively small width of the MAB shelf relative to the North Sea, the difference with the global synthesis is unlikely

to be purely due to the size of the North Sea relative to the average shelf (and therefore longer residence time leading to more degradation of terrestrial DOC). Both the MAB and the North Sea are temperate shelf seas, and a significant proportion of the studies contributing to the Barrón and Duarte (2015) synthesis are sub-tropical or tropical seas. This is consistent with the suggestion of Sharples et al. (2017) that low latitude shelf seas are significant exporters of allochthonous organic matter due to weak Coriolis-driven circulation and greater DOC loading from tropical than temperate rivers. This, however, does not directly explain the greatly enhanced concentration at open ocean salinity implied by the global synthesis; although, all else being equal the sum of autochthonous plus allochthonous DOM would be greater if less allochthonous DOM was degraded. DOC concentrations at high salinity in the North Sea are lower than the values presented by Barrón and Duarte (2015), which potentially implies a considerably smaller net DOC flux per unit area of shelf for the North Sea (and the MAB) than the global average. Nonetheless the evidence from this study strongly suggests that the North Sea is a source of DOC to the open Atlantic, although the strength of this source appears interannually variable. It is not clear from our data or other studies whether the North Sea is a source of sink for open ocean DON, but it appears possible that the direction of net flux might change depending on prevailing conditions.

## 4.3 Differences in carbon inventory between 2011 and 2012

The two summers observed appear to be significantly biogeochemically different, not least in the concentration and C:N ratio of DOM observed, but also in nutrient regime, and physical oceanography. The apparent change in DOC concentration across the whole of the North Sea basin between August 2011 and August 2012 is between 20 and 40 μM. There is little spatial variability in this pattern, with higher DOC in all regions/ depths within this range (Figures 5-9, Table 3). Therefore we are confident in assuming a uniform change throughout the entire North Sea volume of ~42 x$10^3$ km$^3$ (the volume used in the carbon budget for the North Sea by Thomas et al. (2005a)) for the purposes of extrapolating the carbon inventory change. Taking 20 and 40 μM as the lower and upper estimates of interannual difference, this represents a change in the carbon inventory of the North Sea of approximately 10 to 20 Tg (or about 1 to 2 Tmol), or roughly 30 to 40% change in the DOC inventory on the basis of our observations.

This compares to the strength of the DIC enrichment pump for the North Sea estimated by Thomas et al. (2005a) of ~30 Tg C yr$^{-1}$. This enrichment pump represents the accumulation of DIC in the bottom waters of the northern North Sea as a result of POC export from the surface. Could it be possible that the interannual difference in DOC observed is simply a difference in the partitioning of carbon between the dissolved inorganic and organic forms in the two years? If so, we might expect to see higher DIC in the 2012 than in 2011 by a similar magnitude to the DOC change (20-40 μM). DIC data collected as part of the UK Ocean Acidification programme (Naomi Greenwood pers. comm.) from the same cruises as our summer data shows no significant difference in concentration between the two years (t-test, $p < 0.05$), so the change we observe is not likely to be due to a change in partitioning between dissolved inorganic and organic pools.

We do not have POC measurements for summer 2011, so cannot compare the particulate organic carbon pool between the two years. However, as POC in 2012 had an average concentration of ~2 μM and maximum concentration of 5.9 μM, the largest potential POC change (i.e. from zero in 2011 to 5.9 μM in 2012) cannot possibly account for the lower DOC of 20-40 μM in 2012. We discount the idea of a transfer of carbon between the DOC and particulate inorganic carbon on the basis of no obvious mechanism or direct link between these pools and therefore we conclude that the DOC difference is a real change in water column inventory of total carbon between 2011 and 2012.

Spread over the entire surface area of the North Sea the difference in DOC inventory would represent a carbon flux of between 1.5 and 3 mol m$^{-2}$, which is the same order of magnitude as the best estimates for annual net $CO_2$ uptake by the northern North Sea (Thomas et al., 2005b; Wakelin et al., 2012). This change in DOC inventory is, therefore, potentially an important change in the carbon budget of the North Sea, at least for the years concerned. Depending on the source and fate of the DOM, this could represent a substantial change in the magnitude of air-sea $CO_2$ flux and/or carbon export form the North Sea to the deep ocean between these two years, and is indicative of a system whose carbon cycling demonstrates considerable interannual variability. It is therefore important that we try to understand why this might have occurred and we offer some possible explanations below.

### 4.4 Why are the two summers so different?

The difference between summer 2011 and summer 2012 could be due to within-year factors. Marginally lower temperatures in 2011 might have inhibited the remineralisation of DOM. Higher primary productivity might have led to greater DOM production (nutrient levels in northern bottom waters suggest a greater inorganic nutrient inventory to drive spring phytoplankton growth and temperatures indicate weaker stratification in 2011); and subsequently stronger N limitation relative to P (as suggested by higher levels of unutilised P in surface waters in 2011) may have caused greater overproduction of carbon-rich DOM. Greater release from sediment DOM pools is also possible and would be consistent with elevated ammonium concentrations in well-mixed southern waters in 2011.

Given that the largest differences in DOC concentration occur at the highest salinities (Figure 6), where ocean influence is greatest, it is likely that there is a strong physical driver for the difference. DOC concentrations observed in winter 2011/12 in shallow coastal stations mostly in the southern part of the North Sea are among the highest observed in this study, of up to and above 100 μM. These observations also coincide with some of the highest salinities observed in the study (comparable to deep northern North Sea bottom water, Table 2). The winter salinity distribution (Figure 11a) shows highest salinity waters to the south (presumably inflow from the English Channel) and a northward gradient of increasing salinity from low salinities around the UK coast north of 52 °North. The higher salinities to the north of the winter survey areas were associated with the lowest DOC concentrations observed during the cruise (Figure 11b). Overall these data indicate very high DOC, high salinity waters

flowing in to the southern North Sea via the English Channel and low DOC (extrapolated open ocean endmember of ~50 µM) to the North, mixing with the river outflows along the east coast of the UK. This is different to the observations of the previous summer, where high salinity northern bottom waters had DOC concentrations of about 70 µM (Table 3). This may indicate a flushing and renewal of North Sea waters with N. Atlantic water from the North, whilst the high DOC (75-150 µM), high salinity water flowing in from the South, possibly represents the remnants of summer waters being pushed into the North Sea from the Channel and Western Approaches. If this is so, then the winter exchange in 2011/12 may have been responsible for a significant delivery of DOC to North Atlantic as the shelf system was flushed.

Winter 2011/12 had a strongly positive NAO index (Bai et al., 2014; Hurrell, 2017), consistent with a strong shelf-edge exchange and flushing circulation of the North Sea basin (e.g. Salt et al., 2013; Sheehan et al., 2017). Winther and Johannessen (2006) find that short periods of positive NAO can enhance on-flow of surface water onto the shelf in shallow regions such as the Malin shelf, but sustained periods of NAO forcing are required to drive onwelling of deep water through the Norwegian trench. Given that the DOC concentrations in 2012 were consistent with the deep North Atlantic endmember, it is quite plausible that such onwelling may have occurred over winter 2011/12 during a period of intensely positive NAO.

The preceding three years were a period of strongly negative NAO index (Bai et al., 2014; Hurrell, 2017), suggesting relatively little exchange between the waters of the NW European Shelf and the North Atlantic over this period and small output through the Norwegian trench. Elevated DOC concentrations throughout the survey in 2011 might then represent the accumulation of DOC over multiple years of productivity, which was then mixed off-shelf in the winter of 2012; explaining substantial enhancement of DOC concentrations over the predicted conservative mixing line in summer 2011. In a period of high shelf-edge exchange associated with NAO this water would be refreshed with water more consistent with the North Atlantic endmember, resulting in a net export of DOC of approximately the magnitude of the annual $CO_2$ uptake. Without the full northerly extent of the survey in 2012, however, it is not possible to resolve these details conclusively and further years of summer data are needed to resolve the interannual variability and stoichiometry of DOM processing in the North Sea. However this mechanism is strongly consistent with the conclusions of a recent study of the Celtic sea, which concludes that differences between observations of DIC and inorganic nutrients between 2014 and 2015 can be explained by a flushing event which exported carbon-rich organic matter which had potentially accumulated over multiple years from the shelf and simultaneously imported new nutrients onto the shelf to reset the system (Humphreys et al., 2018). As such our data adds further weight to the concept that irregular flushing of carbon-rich organic matter off the shelf is a potentially key element of the continental shelf pump, at least on the NW European shelf.

**4.5 C:N stoichiometry of DOM**

As identified by Humphreys et al. (2018), the stoichiometry of the organic matter pool is potentially important for the efficiency of carbon uptake on the shelf and subsequent export. The C:N ratio in all surface waters in summer in both years of our study has a rather similar average value of (11.3 ± 1.4 and 11.7 ± 2.3) (Table 3), which is consistent with surface DOM production as previously observed in other studies in the North Sea (Van Engeland et al., 2010) and other continental shelf waters (Bates and Hansell, 1999; Hansell et al., 1993; Hopkinson et al., 1997, 2002; Kim and Kim, 2013; Wetz et al., 2008). The bottom water C:N changes from being higher than surface values in 2011 (13.4 ± 3.6), to being lower than surface values in 2012 (9.3 ± 1.8). DOM is therefore depleted in nitrogen relative to primary production in both years (note the POC:PON in 2012 of about six to seven, as expected for primary production), but DOM in bottom waters is N-poor in 2011 and N-rich in 2012 relative to surface waters. This is potentially due to the greater breakdown of POM to yield DOM with lower C:N in 2012, but this cannot explain the strong trend of increasing DON with salinity in northern stratified bottom waters.

The change in C:N ratio with salinity (Figure 6e and f, Figure 9d) reveals striking differences between 2011 and 2012, with C:N increasing with salinity in 2011 to values above the expected North Atlantic endmember of 11-16 (see end member synthesis in Table S2 of Supplementary Information), particularly seen in bottom waters of the stratified northern North Sea. In 2012, however, the C:N ratio decreases to below 10 at high salinities in bottom waters, due to the combination of higher DON and lower DOC than in the previous year. This signal is consistent with the deep North Atlantic endmember, although there is large variability in the literature in observations of deep North Atlantic DON, from 2.7 to 7.7 µM so the range of potentially consistent values is correspondingly large. DON concentrations at the higher end of this range are seen at 35 °N in the open North Atlantic by Kähler and Koeve (2001), but also in deep waters of the Faroe-Shetland channel, associated with Norwegian Sea / Arctic Ocean outflow by Kramer et al. (2005). Lower DON concentrations are observed offshore in the bay of Biscay by Aminot and Kérouel (2004) and in the open North Atlantic between Greenland and Portugal by (Álvarez-Salgado et al., 2013). In the case of northern, high salinity waters in this study, relatively high DON of 6 to 7 µM at high salinity in 2012 is in good agreement with deep waters from the Norwegian Sea/ Arctic Ocean outflow, which would likely be advected onto the shelf if deep Atlantic waters were advected to the North Sea via the Norwegian trench under strong NAO winter conditions, as suggested by Winther and Johannessen (2006). The low C:N values in 2012 are thus consistent with on-flow of water with low C:N DOM. As such, it is possible that water exchange with the Atlantic between 2011 and 2012 led to a net export of DOC from and a net import of DON into the North Sea.

The significant difference between the northern bottom waters in the two years is further explored in Figure 12, which compares DOM stoichiometry in the northern bottom waters in 2011 and 2012 with literature values of the subsurface North Atlantic end member, representing relatively old water not expected to show interannual variability (e.g. Hopkinson and Vallino, 2005). We compare the location of samples in DOC–DON space with possible Atlantic end members from the studies synthesised in Table S2. In 2011, high salinities tended to be associated with lowest DOC and DON concentrations, and all high salinity values can be seen to be very inconsistent with end member data, with elevated DOC at all DON concentrations

relative to the end member ranges. In 2012, high salinities were associated with some of the highest concentrations of DON observed in northern bottom waters, and these high salinity, low DOC (<50 µM) waters with DON concentrations of 6-7 are strongly consistent with the deep Atlantic background concentration observed by Kähler and Koeve (2001) and somewhat consistent with the deep waters of Arctic Ocean origin observed by Kramer et al. (2005), although with lower DOC. We

therefore conclude that the northern bottom waters in 2012 are likely to be strongly influenced by the on-flow of new, deep water of Atlantic or Arctic Ocean origin, supporting our hypothesis that flushing with new, deep water in winter 2011/12 substantially changed DOM composition and concentration in the North Sea between the two summers.

## 5. Conclusion

The data presented in this paper demonstrate that there is significant complexity in the DOM dynamics in the North Sea both spatially and inter-annually, although there is a strong spatial gradient of decreasing DOC and DON concentrations with distance from land and with decreasing salinity. This gradient may arise in the North Sea from estuarine and coastal DOM production by marine phytoplankton or represent the transport and processing of terrestrially-derived material in the coastal zone. The apparent freshwater end members are consistent with either a fluvial (i.e. allochthonous) source or near-coastal

autochthonous production driven by riverine nutrients. Data from another study (Painter et al. 2018) suggests that the contribution of terrestrial sources to the DOM pool in the open North Sea is small, so we suggest that nutrient-driven autochthonous production is the more likely explanation. The analysis of DOM/salinity gradients here is constrained by our poor understanding of the lifetime of DOM in the North Sea, and of the source of high DOM in near-coastal waters (autochthonous, nutrient-driven vs riverine source). More measurements of DOC/N in rivers and estuaries flowing into the

North Sea, particularly transects from low to high salinity would be a valuable addition to our knowledge of low salinity DOM cycling and production in this region.

Overall, DOC concentrations are lower in the North Sea than average values in the global synthesis of shelf DOC concentrations by Barrón and Duarte (2015). Long residence times in the North Sea, compared to the global shelf average may

help to explain the lower-than-average DOC concentrations, with little or no terrestrial DOM reaching the shelf break (e.g. Painter et al., 2018). Longer residence time also means longer for the production and processing of autochthonous organic matter, so the oldest, most saline waters of the northern North Sea may provide a good synthesis of shelf processes or even some 'memory' of multi-year organic matter processing signals during periods of low exchange with the Atlantic.

The spatial pattern (i.e. dominated by mixing) is similar in both years but the actual concentrations, particularly of DOC, are different. High interannual variability appears to dominate DOC/DON, particularly in the deep bottom waters in the northern North Sea, which are those that are most likely to be leaving the shelf the following winter. The source of elevated DOC

concentrations in 2011 is potentially the same as the high DOC signal seen by Thomas et al. (2005a), in their study attributed to onwelling water; but we establish that at least in 2011 the elevated DOC is inconsistent with the open ocean DOC concentration, suggesting an on-shelf source (Figure 9), and in disagreement with the budget of Thomas et al. (2005a) which suggests net respiration of ocean DOC in the North Sea overall. Simultaneous measurements of DON sheds more light on this

than observations of DOC alone: in 2011 at least, high DOC in deep waters appears associated with high C:N ratio (i.e. depleted DON). This is consistent with in-situ production of high DOC organics, and/or the preferential remineralisation of N over C from DOM. Either way this high C:N DOM is a potentially important component of the shelf carbon pump, allowing greater carbon uptake for a given winter stock of inorganic nutrient than for non-carbon enriched (i.e. Redfieldian) DOM production. The differences between 2011 and 2012 described here suggest important interannual differences in the scale of shelf sea

carbon export in this region, and perhaps at other shelf sea boundaries. We note that the DOC concentrations observed by Painter et al. (2018) in summer 2016 in the North Sea are comparable in magnitude and distribution to those in 2011, suggesting that the apparent accumulation of DOC in the North Sea may be a regular occurrence. Further years of data and greater seasonal detail are needed to quantify this effect and understand the complex interaction roles of mixing, benthic interaction and nutrient limitation in the system.

The data and subsequent analysis we present here leads us to hypothesise that intermittent flushing of the North Sea may be a key driver of both the variability in the carbon inventory and also the strength of the shelf pump for carbon in the region, and possibly elsewhere. Periods of low exchange with the open ocean are associated with reduced nutrient inputs to the system and would be characterised by high C:N dissolved organic matter, which represents an accumulation of carbon in the system

and thus continued uptake of atmospheric $CO_2$ in spite of reduced carbon export to the deep ocean via circulation. This high C:N DOM could accumulate via preferential remineralisation of DOM nitrogen over carbon and/or nutrient stress leading to overflow production of carbon rich organics by phytoplankton (e.g. Humphreys et al., 2018; Lønborg et al., 2010; Prowe et al., 2012). Either of these mechanisms are likely to be compounded by increased periods of relative isolation of shelf waters from the open ocean, when nutrient inputs are reduced and the total nitrogen inventory is likely to decrease due to loss from

denitrification in sediments. Our data is strongly supportive of this hypothesis, providing both circumstantial evidence of a flushing event based on winter salinity data and comparison with open ocean endmembers and evidence of elevated DOC concentrations, low DON and high C:N in the high-salinity shelf waters in summer 2011 prior to the proposed winter flushing event. The change in the DOC inventory observed here suggests that DOC may represent the most variable pool of carbon in the shelf water column and therefore the biggest control on annual $CO_2$ uptake and subsequent off-shelf export. Greater

understanding of the dynamics of DOM production, degradation and stoichiometry is needed. In particular, the interacting roles of sediment-water interactions, nutrient limitation and variable water residence times/ flushing events needs to be better understood. This is an important area of future study in better quantifying and predicting the role of shelf seas in carbon uptake and export to the deep ocean.

**Acknowledgements**

S. Chaichana was supported in her studies by a scholarship from the The Royal Thai Government. M. Johnson was supported by UK Natural Environment Research Council (NERC) grant NE/K00168X/1 (Data Synthesis and Management of Marine and Coastal Carbon), part of the NERC Shelf Seas Biogeochemistry Programme. We thank the scientists and crew of the RV

5  Endeavour from the Centre for Environment, Fisheries and Aquaculture Sciences (Cefas) in Lowestoft, UK for their help and support during cruises. All authors acknowledge the support of the Cefas-UEA joint Centre for the Sustainable Use of the Seas (CCSUS) which facilitates collaboration between the two institutions.  The full dataset of DOM concentrations, temperature, salinity and nutrients will be made openly available via publication in a data centre with a DOI assigned, prior to final publication. Two anonymous reviewers helped greatly in improving this manuscript.

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

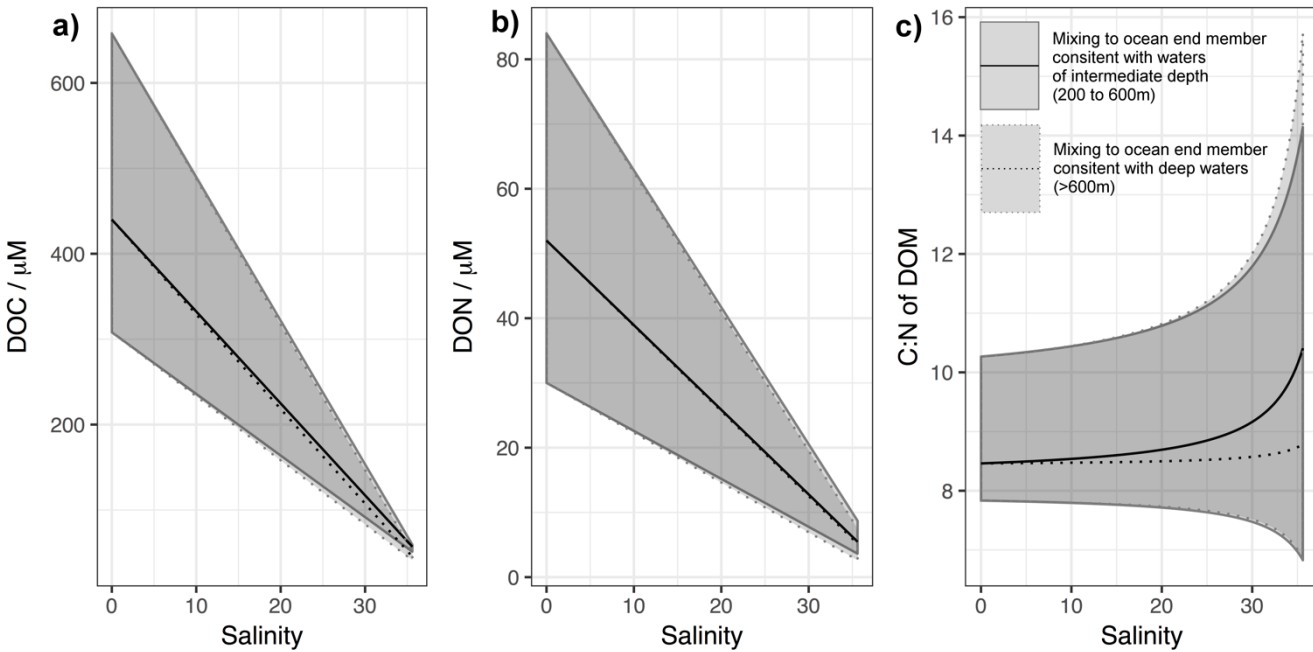

**Figure 1. Predicted variation of a) DOC, b) DON and c) C:N of DOM with Salinity based on literature values of observed end member concentration ranges for North Sea and Baltic rivers and the open Atlantic adjacent to the NW European shelf. Data and originating studies detailed in Tables S1 and S2 of the supplementary material. Two ocean end member ranges are presented, one for intermediate waters (200 – 600 m) and one for deep waters (>600 m). The C:N values presented here are intended to demonstrate the likely trend with salinity. C:N of DOM presented represents only a subset of all possible values: upper (lower) estimates of DOC are divided by upper (lower) estimates of DON to give upper (lower) estimates of C:N. Including C:N predicted by upper DOC divided by lower DON and vice versa gives an unrealistically large range of values.**

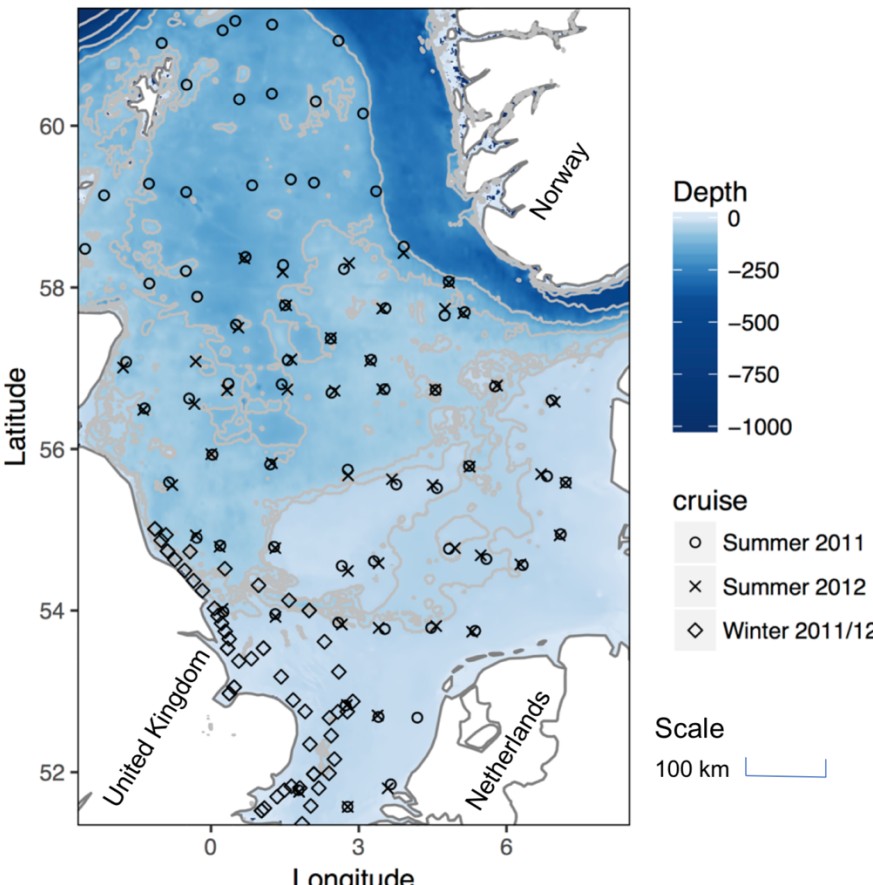

Figure 2. The North Sea study area, showing sampling points from each of the three cruises (note that the two summer cruises occupy notionally the same stations, each in separate ICES grid square, each year hence the close overlap). Also shown is bathymetry, with 40 m, 50 m, 100 m, 200 m, 400 m, 600 m and 1000 m contours shown in light grey. Note that the division between well mixed southern waters and the seasonally stratifying norther region follows the bathymetry, approximately at the 40 m contour running from around 54 N on the UK coast to approximately 57 N on the Danish coast.

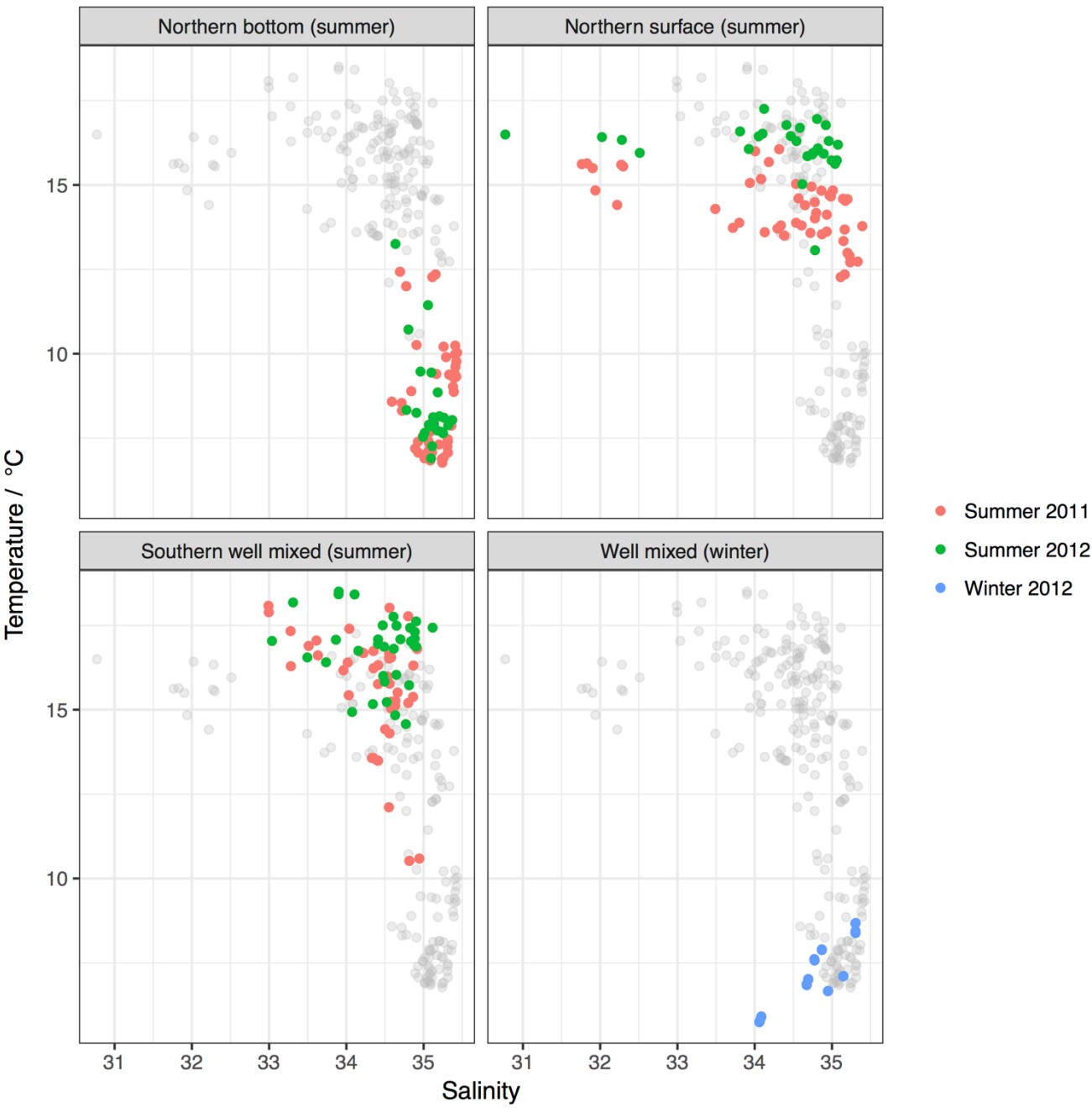

**Figure 3. Temperature-salinity plots divided by location/ season and cruise. Grey points show the whole data set from all other water types / seasons.**

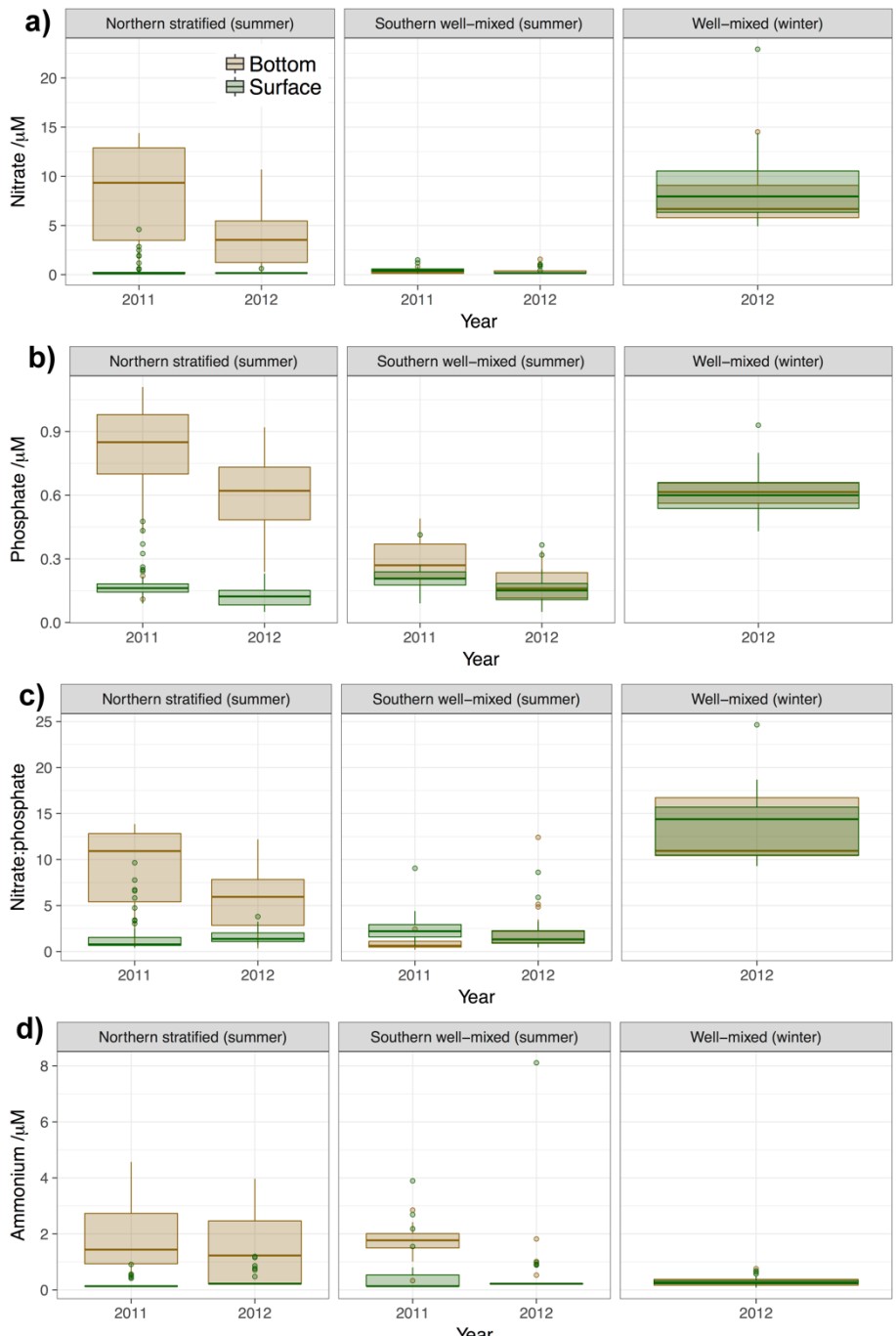

**Figure 4. Summary of inorganic nutrient observations demonstrating the differences between surface and bottom samples, different mixing regimes, seasons and years. a) nitrate, b) phosphate, c) nitrate to phosphate ratio and d) ammonium concentration. Box and whisker plots show statistical summary of the data where the thick horizonal like represents the median value, the extent of the boxes represents the interquartile range and whiskers represent the full range of data up to 1.5 interquartile distances from the median. Any points outside this range are shown as discrete points.**

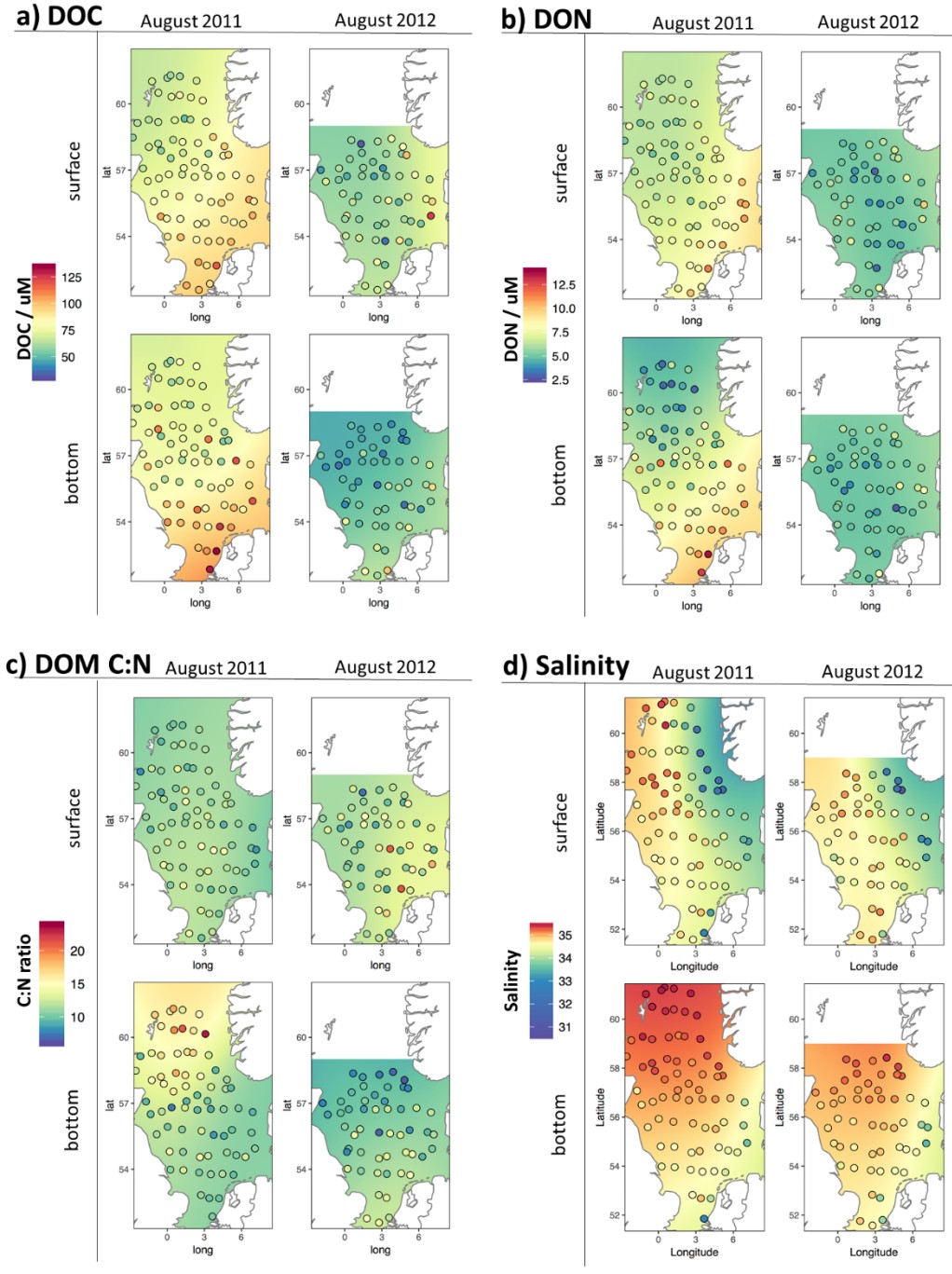

**Figure 5. Distribution of a) DOC, b) DON, c) C:N ratio of DOM and d) salinity for surface and bottom waters in summer 2011-2012. Points represent discrete observations and the underplotted colour gradient is a 2-dimensional Gaussian smooth of the discrete data to demonstrate the spatial trend.**

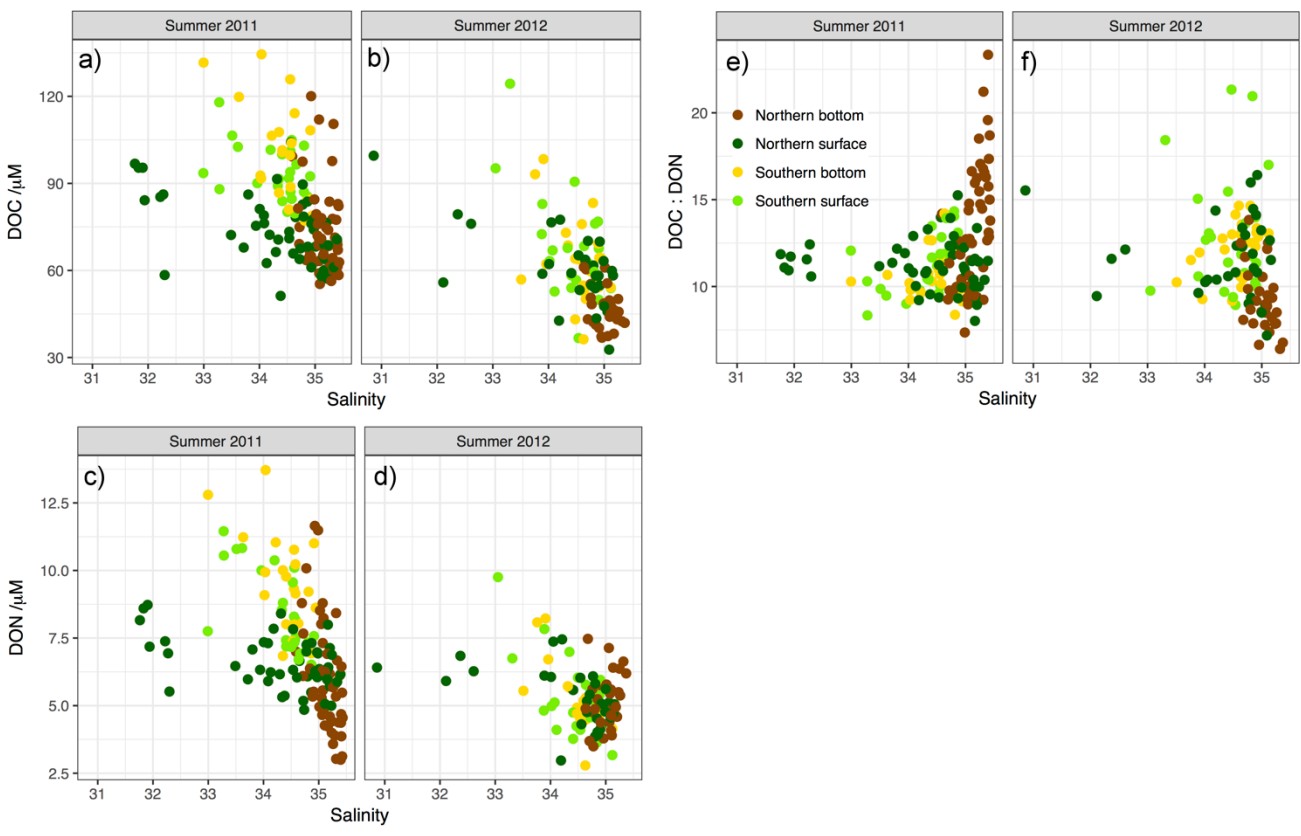

**Figure 6. Property-salinity plots for a) DOC, c) DON and e) C:N ratio of DOM for summer 2011. Corresponding plots for summer 2012 are shown in panels b), d) and f). Note that regression analysis for property-salinity relationships is presented in Table 4.**

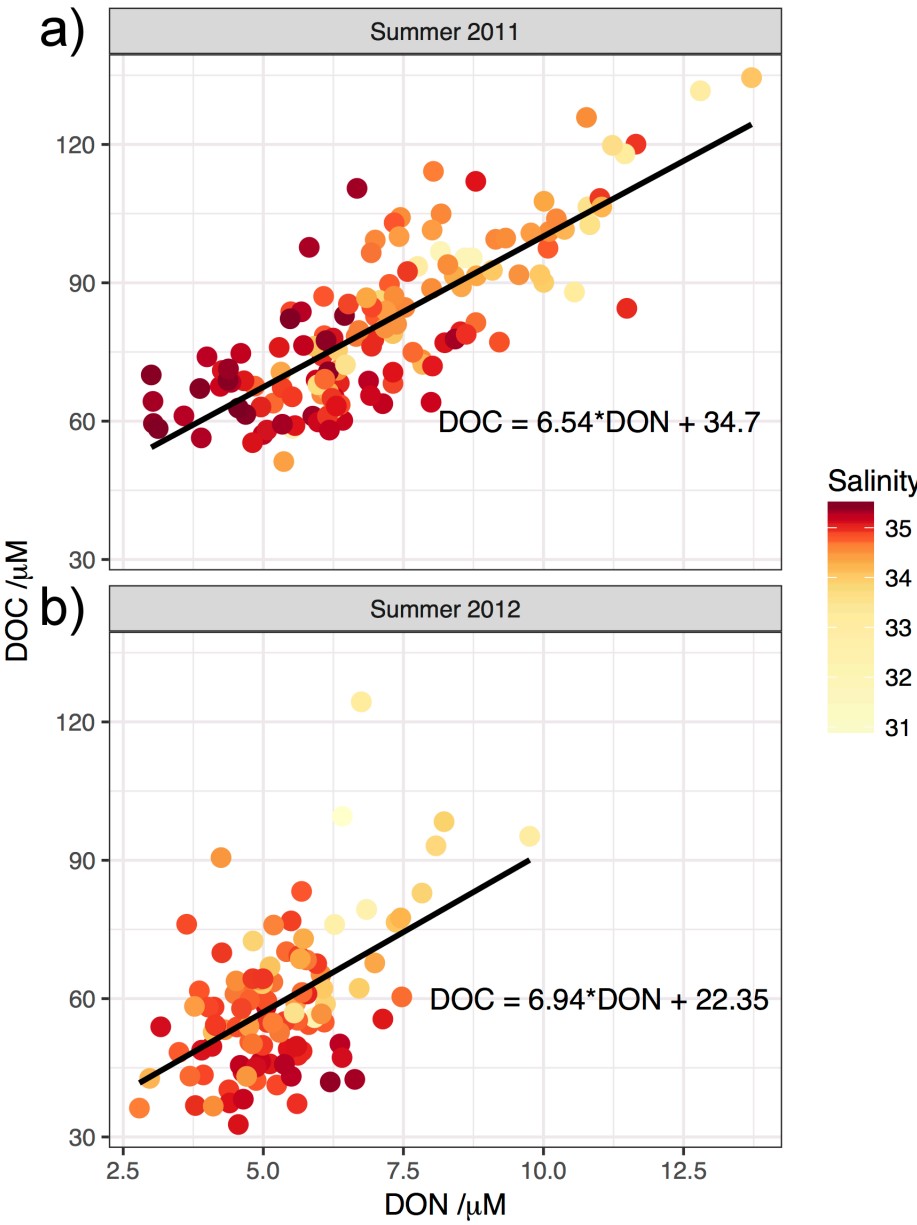

**Figure 10. Element-element plots of DOC vs DON for summer surveys a) August 2011; b) August 2012. Points are coloured by salinity. Lines represent application of standard linear least squares regression models to the relevant data; equations for which are quoted on the plots.**

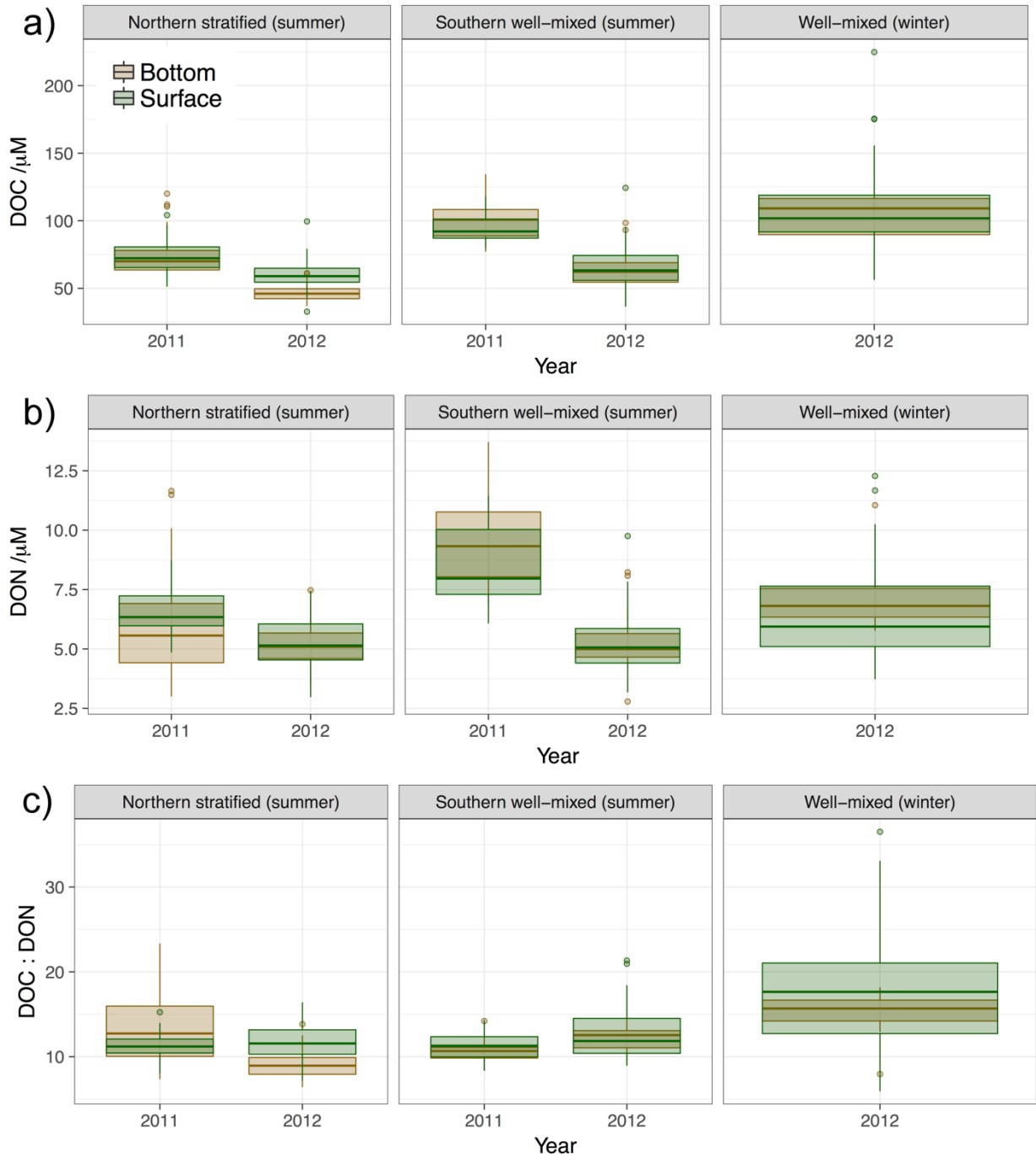

**Figure 7. Summary of DOM concentrations a) DOC, b) DON and c) DOM C:N ratio. Box and whisker plots show statistical summary of the data where the thick horizonal like represents the median value, the extent of the boxes represents the interquartile range and whiskers represent the full range of data to 1.5 interquartile distances from the median. Any points outside this range are shown as discrete points.**

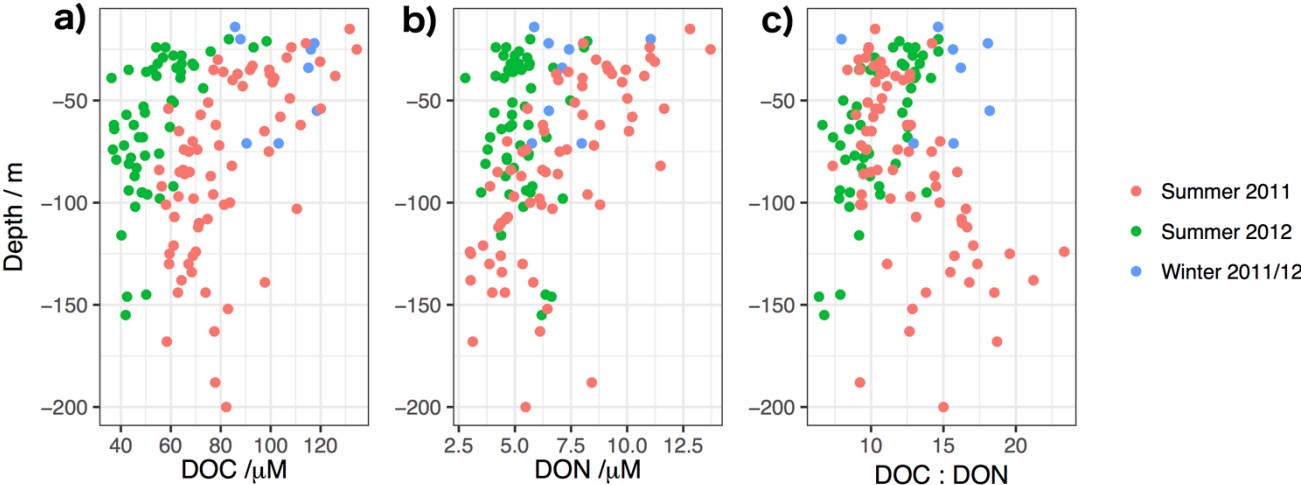

Figure 8. DOC (a), DON (b) and DOM carbon to nitrogen ratios (c) in bottom water samples, plotted against sampling depth (approximately 10 m less than total water column depth). Red points from summer 2011, green points from summer 2012, blue from the intervening winter cruise.

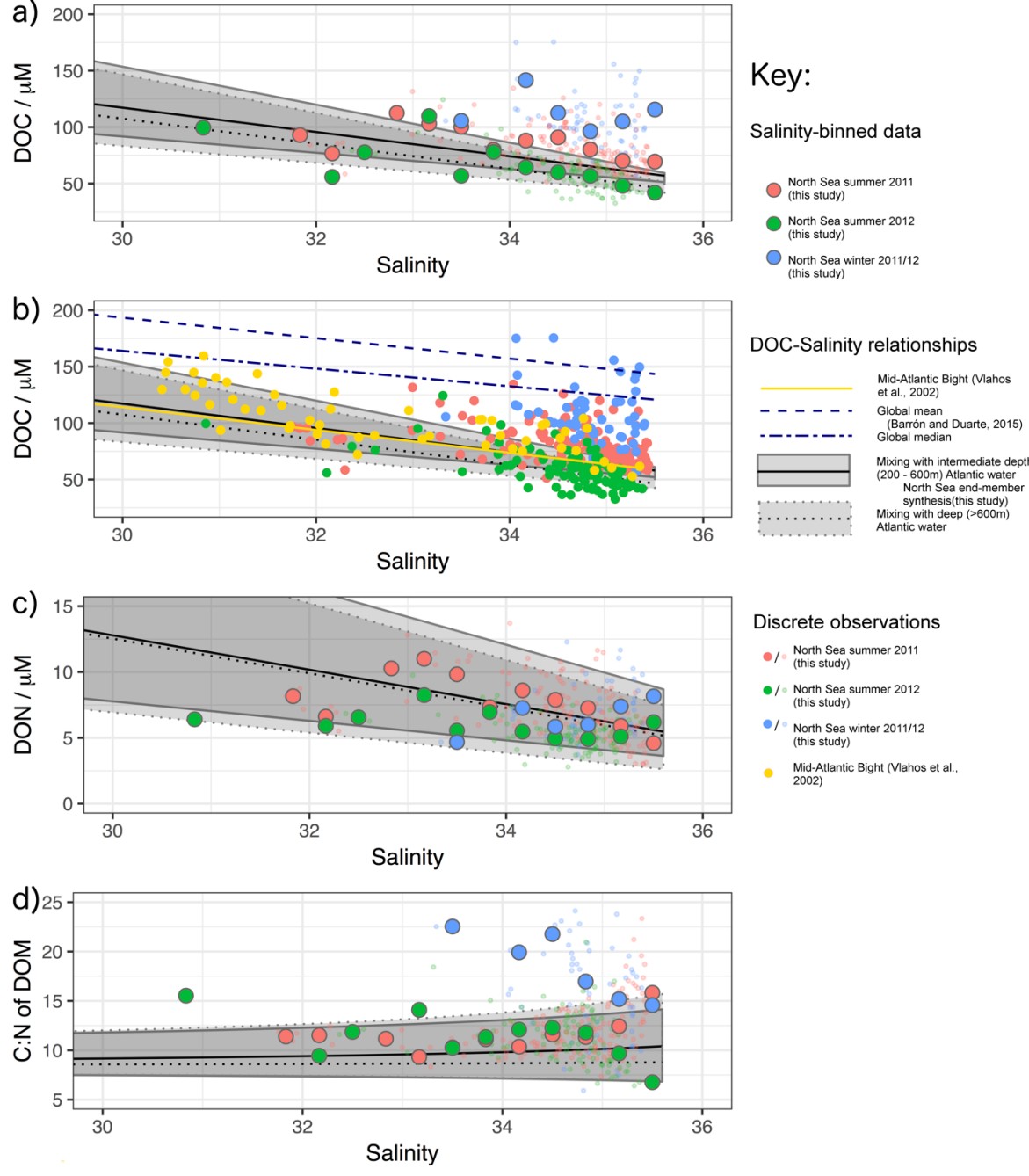

**Figure 9. Comparison of a) DOC, c) DON and d) C:N relationships with salinity, compared to conservative mixing predictions.** Small points represent discrete observations and large points are binned averages spaced at 1/3 of a salinity unit. The solid and dotted grey lines/areas represent predicted conservative mixing lines between rivers and the shallow and deeper Atlantic end members, respectively. In panel b) DOC data form this study is compared to the data (black dots) and salinity relationship (dot-dash line) of Vlahos et al. (2002) in the Mid-Atlantic Bight, as well as the mean (long dash) and median (long dash – short dash) relationships from the global synthesis of Barrón and Duarte (2015).

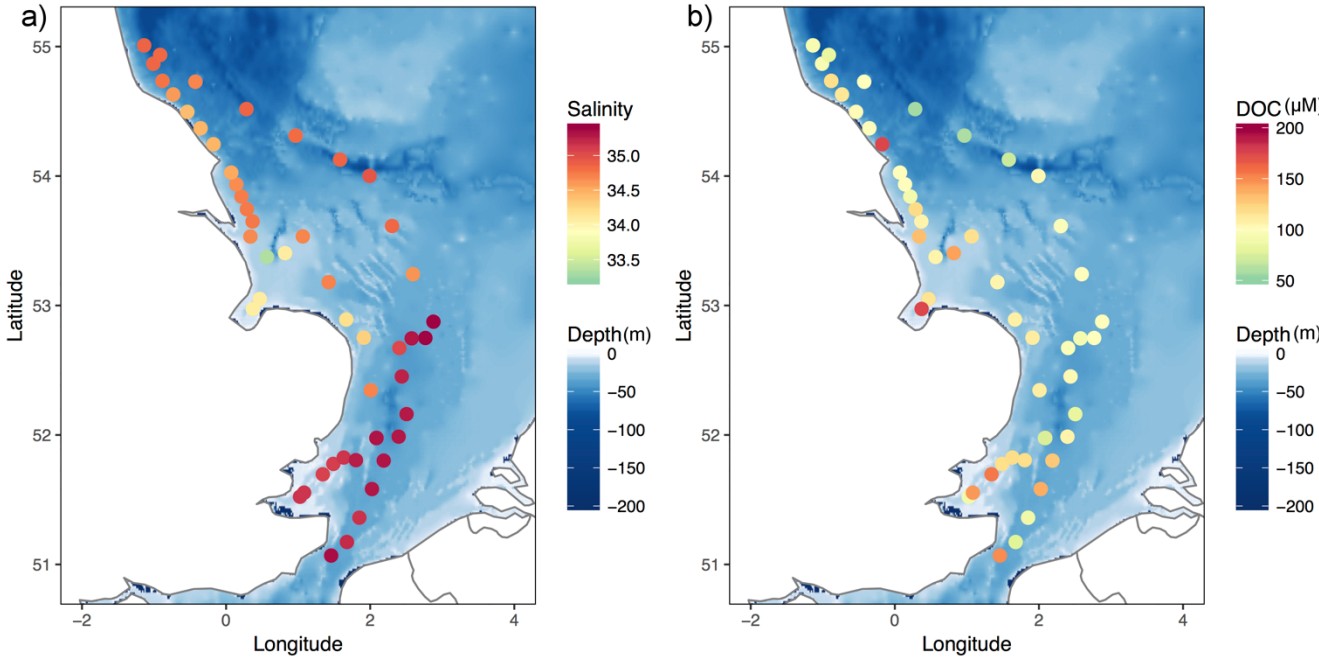

**Figure 11. Salinity (a) and DOC (b) distributions observed during the winter cruise in January 2012.**

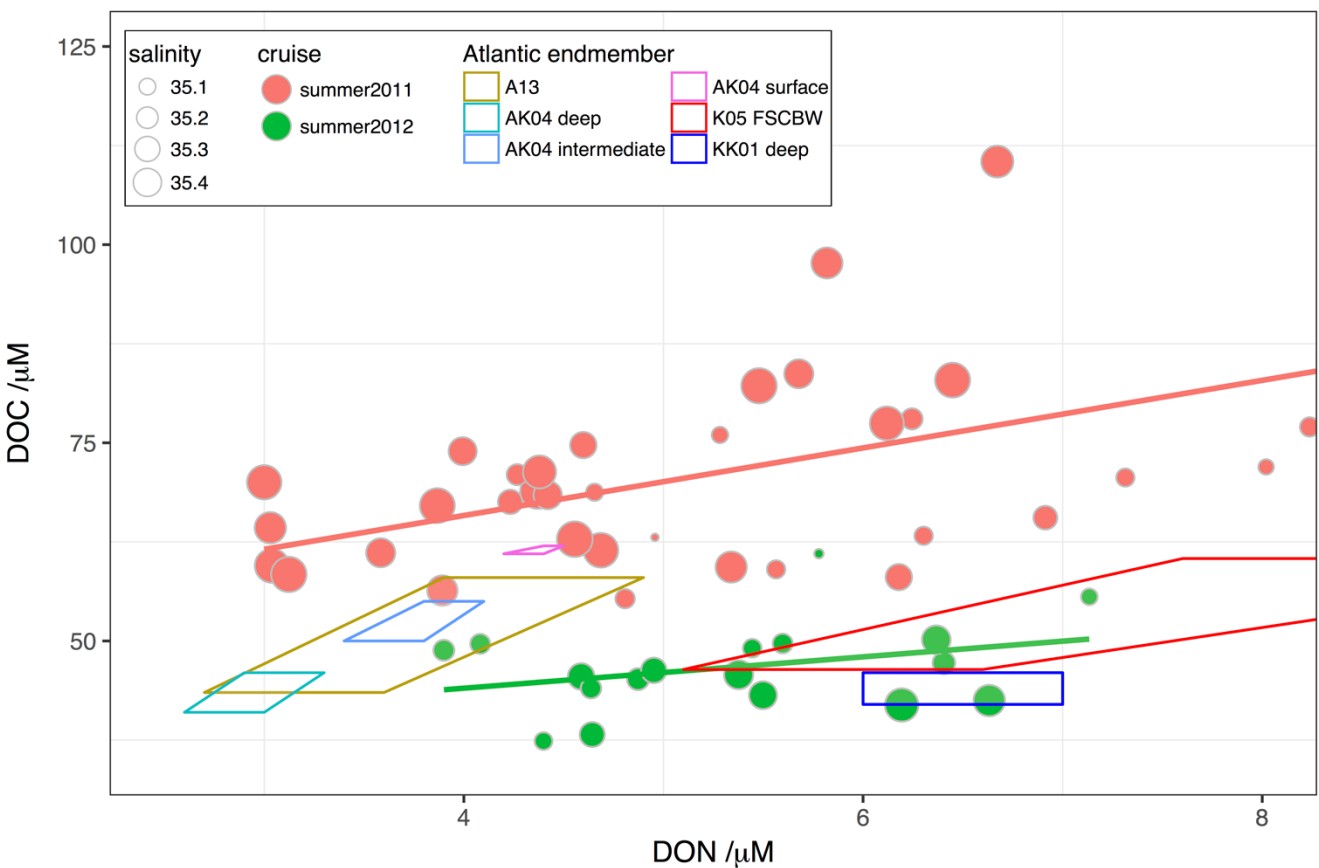

**Figure 12. DOM stoichiometry from northern stratified bottom water samples in summer 2011 (red) and 2012 (green), compared with open North Atlantic concentrations as follows. A13: Álvarez-Salgado et al. (2013); AK04: Aminot and Kérouel (2004); K05: Kramer et al. (2005); KK01: Kähler and Koeve (2001). These 'end member ranges' were constructed in DOC/DON space based on the ranges of concentrations of DOC and DON quoted in the relevant papers, constrained by limiting possible combinations of DOC and DON that were consistent with the quoted range of observed C:N ratios (in the absence of the full dataset). The full range of DOC concentrations observed in the studies is covered by these derived end-members, and the large majority of the DON concentrations.**

**Table 1. Summary of sampling cruises in the North Sea.**

| Cruise number | Dates | Season |
|---|---|---|
| CEND 14/11 | 8 Aug – 7 Sep 2011 | Summer 2011 |
| CEND 02/12 | 20 Jan – 31 Jan 2012 | Winter 2011 |
| CEND 13/12 | 9 Aug – 23 Aug 2012 | Summer 2012 |

CEND is abbreviation for the research vessel Cefas Endeavour in its cruise naming nomenclature.

**Table 2. Summary of physical and inorganic nutrient parameters during this study.**

| | Temperature (°C) | Salinity | DIN [a] (µM) | DIP [b] (µM) | DISi [c] (µM) |
|---|---|---|---|---|---|
| **August 2011** | | | | | |
| Southern well-mixed | 15.5 ± 1.7 | 34.3 ± 0.5 | 1.5 ± 1.1 | 0.2 ± 0.1 | 2.2 ± 1.5 |
| | (10.4-17.8) | (33.0-34.9) | (0.2-4.7) | (0.1-0.5) | (0.4-7.7) |
| Northern stratified surface | 14.2 ± 1.0 | 34.3 ± 1.0 | 0.7 ± 1.1 | 0.2 ± 0.1 | 1.3 ± 0.6 |
| | (12.2-16.0) | (31.8-35.4) | (0.2-4.7) | (0.1-0.5) | (0.5-3.7) |
| Northern stratified bottom | 8.4 ± 1.6 | 35.1 ± 0.2 | 9.9 ± 4.4 | 0.8 ± 0.3 | 4.5 ± 1.6 |
| | (6.7-12.4) | (34.6-35.4) | (0.9-16.2) | (0.1-1.1) | (0.3-6.8) |
| **January 2012** | | | | | |
| Well-mixed | 7.2 ± 0.9 | 34.8 ± 0.4 | 8.8 ± 3.3 | 0.6 ± 0.1 | 5.4 ± 1.0 |
| | (5.7-8.7) | (33.3-35.4) | (5.3-23.0) | (0.4-0.9) | (4.1-8.9) |
| **August 2012** | | | | | |
| Southern well-mixed | 16.2 ± 1.7 | 34.5 ± 0.5 | 0.8 ± 1.2 | 0.2 ± 0.1 | 1.5 ± 1.0 |
| | (11.0-18.5) | (33.1-35.1) | (0.4-8.3) | (0.1-0.4) | (0.3-5.2) |
| Northern Surface | 16.3 ± 0.8 | 34.3 ± 1.0 | 0.5 ± 0.3 | 0.1 ± 0.0 | 0.9 ± 0.5 |
| | (13.2-17.4) | (30.9-35.2) | (0.4-1.5) | (<LOD-0.2) | (0.1-1.6) |
| Northern Bottom | 8.8 ± 0.6 | 35.0 ± 0.2 | 5.5 ± 3.0 | 0.6 ± 0.2 | 3.2 ± 0.8 |
| | (7.5-10.5) | (34.6-35.4) | (0.4-11.2) | (0.2-0.9) | (1.2-5.1) |

Mean values are presented in mean ± SD, SD is standard deviation. Range values are showed in parentheses. Limit of detection (LOD) is 0.1 µM for phosphate in August 2012 samples. For parameters presented < LOD, the half of detection limit was used to calculate the mean value.

[a] DIN = the sum of nitrogen concentration in the form of nitrate (total nitrate plus nitrite) and ammonium

[b] DIP = phosphate concentration

[c] DSi = silicate concentration

**Table 3. Summary of dissolved organic carbon and nitrogen observations in this study.**

| | DOC (μM) | DON (μM) | DOC:DON | POC (μM) | PON (μM) | POC:PON |
|---|---|---|---|---|---|---|
| **August 2011** | | | | | | |
| Southern well-mixed | 97.5 ± 13.7 | 9.0 ± 1.8 | 11.1 ± 1.6 | - | - | - |
| | (77.1-134.5) | (6.1-13.7) | (8.3-14.3) | | | |
| Northern Surface | 73.8 ± 11.6 | 6.6 ± 1.0 | 11.3 ± 1.4 | - | - | - |
| | (51.2-104.2) | (4.8-8.7) | (8.0-15.3) | | | |
| Northern Bottom | 73.8 ± 14.7 | 5.9 ± 2.1 | 13.4 ± 3.6 | - | - | - |
| | (53.3-120.1) | (3.0-11.7) | (7.4-23.3) | | | |
| **January 2012** | | | | | | |
| Well-mixed | 107.5 ± 29.6 | 6.7 ± 2.0 | 17.3 ± 6.2 | - | - | - |
| | (56.2-224.8) | (3.7-12.3) | (5.9-36.5) | | | |
| **August 2012** | | | | | | |
| Southern well-mixed | 65.5 ± 16.4 | 5.3 ± 1.3 | 12.6 ± 2.8 | 16.0 ± 9.3 | 2.2 ± 1.3 | 7.7 ± 1.8 |
| | (36.3-124.4) | (2.8-9.8) | (8.9-21.3) | (5.8-43.8) | (0.6-5.9) | (5.0-13.2) |
| Northern Surface | 60.7 ± 13.0 | 5.3 ± 1.1 | 11.7 ± 2.3 | 10.5 ± 4.2 | 2.0 ± 0.7 | 5.9 ± 3.1 |
| | (32.7-99.5) | (3.0-7.5) | (7.2-16.4) | (2.7-21.8) | (0.6-2.9) | (1.1-14.2) |
| Northern Bottom | 46.9 ± 6.8 | 5.2 ± 1.0 | 9.3 ± 1.8 | 7.3 ± 3.5 | 1.5 ± 0.7 | 6.0 ± 3.9 |
| | (36.8-61.2) | (3.5-7.5) | (6.4-13.8) | (1.1-16.2) | (0.3-2.7) | (0.7-16.8) |

Mean values are presented in mean ± SD. Range values are showed in parentheses.

**Table 4. Regression analysis of DOC, DON, POC and PON with salinity in each water mass. Note only significant correlation (at the 0.05 confidence level) is presented.**

| Parameters | Surveys | Water mass [a] | R-square ($R^2$) | Slope | $y$-Intercept ± uncertainty | n [b] |
|---|---|---|---|---|---|---|
| DOC | Summer 2011 | NS | 0.2594 | -5.7 | 270.1 ± 47.9 | 50 |
|  |  | SM | 0.1661 | -11.1 | 477.7 ± 129.9 | 45 |
|  | Winter 2011 | SM | 0.0830 | -20.0 | 805.5 ± 304.6 | 60 |
|  | Summer 2012 | NS | 0.4062 | -8.1 | 338.3 ± 63.5 | 30 |
|  |  | SM | 0.2871 | -18.7 | 711.0 ± 153.4 | 46 |
| DON | Summer 2011 | NS | 0.1973 | -0.4 | 20.6 ± 4.1 | 50 |
|  |  | NB | 0.3009 | -5.1 | 183.7 ± 39.5 | 49 |
|  |  | SM | 0.3037 | -1.9 | 75.3 ± 15.3 | 45 |
|  | Summer 2012 | NS | 0.2535 | -0.5 | 23.4 ± 5.9 | 30 |
|  |  | SM | 0.4047 | -1.8 | 67.1 ± 11.3 | 46 |
| POC | Summer 2012 | SM | 0.1004 | -6.3 | 233.6 ± 98.2 | 46 |
| PON | Summer 2012 | SM | 0.1469 | -1.0 | 37.8 ± 12.9 | 46 |

[a] Water masses: NS = stratified northern surface water, NB = stratified northern bottom water, SM = southern well-mixed water

5  [b] Number of sample (n)