# Peer review of "Interannual variability in the summer dissolved organic matter inventory of the North Sea: implications for the continental shelf pump"

_Biogeosciences, 2017_

## Referee Comment (RC1) · Anonymous Referee #1 · 3 Nov 2017

**General Comments**

This paper details a multi-year high-resolution sampling of DOM dynamics in the North Sea. The main conclusions of the work suggest that there is high spatial and temporal variability in the total concentrations and C:N ratio of OM over the sampled periods, and that this inter-annual variability has strong implications for overall carbon budgets in the region. The implications of this work for carbon cycling in the region are important, and I found the paper to be generally well written. However, I struggle with the chosen focus on C:N ratios for elucidating DOM dynamics. I felt that novel portion of this work, that is, the elucidation of the impact temporal variability has on the overall carbon budget(s),

was not emphasized enough. In particular, there are a few main facets of the work I feel need further development if they are to be included in the final manuscript:

1) The objective that the C:N ratio is presented to address- and how this is interpreted and discussed. As mentioned in the work, many factors can affect this ratio! If this is the only metric you use to assess DOM dynamics, you need to be really careful. Can you really answer the larger objectives you outline with the hypotheses that you pose? How does the C:N ratio compare to the chla concentrations, nutrients, and previous work in the region on tracking allochthonous vs autochthonous sources of OM? How does your end-member calculations compare to actual end-members from the literature, and OM composition from other work?

2) Tie the POM work in better. How does this compare to the DOM, and what is this impact on the overall carbon budget?

The discussion and conclusions regarding the relationship of DOM with salinity. This relationship has been found to be conservative with mixing of the major water masses in the North-Baltic Seas transition, however major non-conservative processing on DOM does occur in the region (see eg. Osburn and Stedmon, 2011 Marine Chemistry DOI: 10.1016/j.marchem.2011.06.007). I would like to see further development of this, including tying the current work into previous analyses of DOM composition, etc. What are the implications of local variability, in this context?

3) This is perhaps the most important- I feel the discussion on C inventory should take center stage. How does this work advocate for or against high-resolution measurements? How does it revise or promote our understanding of DOC cycling in the region? How does this compare to other regions and/or the global budget? What are the uncertainties with C budgets, and does this paper help to narrow these?

**Specific Comments**

1 Introduction-
The first paragraph starting on line 23, to me, is the motivation for this work. Clearly relate the following discussions, and set-up to the goals of the study relative to this. Be explicit upfront- what do you hope to find with this work, and how is it novel (e.g. lines 25-29)? Some of this is outlined in later the introduction, but the narrative back to the main objectives of the work is lost throughout the rest of the paper. The authors attempt this tie by stating hypotheses and referring to them throughout. This causes the prose to be a bit awkward- and I suggest instead outlining the overlying research outcomes the authors hope the study will answer. As is, the hypotheses are too narrow to the stoichiometry work, and don't really address our gaps in understanding of the temporal variability of the C cycling in the region. Perhaps a figure would help with synthesizing what we know, and what gaps this study addresses? How has this sort of "high-resolution" work refined carbon budgets in other regions? What are the processes affecting atmospheric CO2 draw down in regions such as shelf seas? How does this relate to the global carbon budget? Why is the North Sea an ideal system to study in this regard?

Page 2, Line 6- Is DIC really the only place for long term carbon storage in the ocean? Better tie in why you are looking at DOC, not DIC.

Page 2, Paragraph starting line 13- I would argue that C:N stoichiometry is a very small part of understanding allochthonous vs autochthonous OM sources, and especially reactivity of DOM. You acknowledge some caveats here, but how does more compound-specific work (such as isotopes, biomarkers, etc) compare to C:N ratios (e.g. Kaiser and Benner, 2012 JGR-Oceans DOI: 10.1029/2011JC007141)? Convince the reader that the stoichiometry is an adequate tool for the objective you are outlining, ie. using C:N ratios to understand DOM source and reactivity. As is, I feel that the discussion and implications of the work rely too heavily on this.

Page 3 line 15- Add "climate" in front of cycles

2 Methods-

**BGD**
In general, I feel the methods are well explained and analytically sound. An interesting paper recently came out in EST Letters that I feel the authors could benefit from regarding "DON" calculations- Saunders et al., 2017 DOI: 10.1021/acs.estlett.7b00416

Page 4, lines 21-22- Why did you exclude the riverine-influenced sites? How does this impact your further discussion of sources and end-members and your hypotheses above?

3 Results- The results section includes a bit of interpretation in it (e.g. see paragraph on page 11 lines 11-17) and should be reworked to include only observations of the data.

Do you have the TS profiles? What about other property/property plots?

Page 7 line 21- What do you mean by noisy?

Page 8, lines 8-12- This is confusing, but is an important distinction. Be clear with your comparisons here, and throughout the rest of the manuscript! Perhaps delineating the water bodies by type for comparisons of measurements over time (eg. Open Atlantic water)?

Page 9, lines 10-11- How does the spatial subset data compare?

Page 9, paragraph lines 20-29- I don't understand the point you are trying to make here. Additionally, the DOM-Salinity relationships, while significant, are not very strong (R2

Page 10, line 28- What are the percent differences between these observations (i.e. interannual vs depth)? Is this statistically significant?

Page 11, line 20- "interesting differences"- what are these differences? Be explicit. This paragraph is confusing, perhaps by splitting up the observations into difference sentences would help for a more succinct narrative.

4 Discussion- I feel that much of the discussion should be reworked- I have a hard time following the structure of many of the arguments in the discussion, in particular the DOM-salinity (section 4.1) and the DOM variability (section 4.3) discussions. Are the end-member data robust enough you could perform an actual mixing analysis (similar to the approach in Perdue and Koprivnjak, 2007 Estuarine, Coastal and Shelf Science doi:10.1016/j.ecss.2006.12.021 ; See also the caveats outlined in using C:N ratio to determine terrestrial vs aquatic sources of OM outlined in this work)? How, specifically, does the nutrient data tie into this? Section 4.1- I am missing the connection between the topic sentence and the following discussion. How does the lack of relationship between chla and POM support or refute your hypotheses?

Section 4.2 – This is your most interesting and novel finding. Do you see large spatial variations that might weaken the budget extrapolation? Are there any physical oceanographic work that support the shifts in exchange of water masses that you discuss? I think this section could be split and both paragraphs expanded upon significantly.

Section 4.3- The discussion of potential benthic inputs of OM must be further expanded upon- while this is an interesting hypothesis, the current arguments do not convince me. Do your turbidity or POC data support the nepheloid hypothesis?

5 Conclusions-

Again, I feel the focus here should be on the C budgets more than the DOM dynamics.

**Technical Corrections**

Comments for throughout the manuscript:
Make sure super and subscripts are correct (e.g. page 3 line 5, page 5 line 17). Check sentence structure for flow, spelling, and punctuation. Below are a few (non-exhaustive) examples: Page 3, line 16 a comma is missing after "2007)" as on page 6, line 9 a comma is missing after "(LOD)". Page 12, line 14 is missing a period. Page 16 line 16 missing a "t" in "this".

Check paragraphs for run-on sentences, which confusing the meaning. Eg. Page 11 lines 19-23.

I feel many of the connecting sentences are awkward and should be reworded to flow better. E.g. page 12 lines 2-4: "In this discussion, we consider..." Check that the citations are imported to the text properly (e.g. page 12 line 9). Make sure the nomenclature is used consistently- ie. DOM, DOC, DON.

---

## Referee Comment (RC2) · Anonymous Referee #1 · 3 Nov 2017

In addition to posted comments, I would like to make a few minor suggestions for clarity of the figures.

Figure 1. This should be zoomed out and/or the land masses labeled. A compass-rose would help as well.

Figure 3. I find this figure hard to follow. Perhaps consider a different way to display this data, for example a temporal evolution plot such as those created in ODV?

[Figure]

---

## Referee Comment (RC3) · Anonymous Referee #2 · 30 Jan 2018

Anonymous review of: "Distribution and C/N stoichiometry of dissolved organic matter in the North Sea in summer 2011-12" by Saisiri Chaichana, Tim Jickells, Martin Johnson

The authors present results from three cruises that sampled the North Sea in August 2011/12 and January 2012. In addition to standard temperature and salinity data, nutrient (nitrate + nitrite, ammonium, phosphate, silicate) and dissolved organic matter (DOC, TDN→DON) are also reported, being the focus of the manuscript. Particulate organic matter and chlorophyll collections are also described, however they seem to be of little focus to the manuscript and were not included in the results and discussion sections. Oxygen data, were either not collected or are not reported. Authors primarily consider salinity, nutrient, DOM concentrations and DOC:DON, exploring relationships by cruise, region and surface/bottom samples.

Much of the discussion is superficial, mentioning other/relevant papers without exploring prior results to gain insights and new findings from the reported data. It is mentioned that more data, particularly sampling other months/years, is necessary to complete analysis. As a result, reader is left wondering why the study was published if it is not complete and inconclusive. There is great potential to supplement limited data with satellite (temperature, chlorophyll, and even Aquarius salinity during the study period, however resolution may be too coarse), temperature (and oxygen?) data, and apparently measured but not reported POC/N and chlorophyll data, which should be done. Interestingly, despite the limited data that the authors are working with, an excessive number of figures are included (both in the manuscript and as supplemental). In grand total, 13 figures are included with the manuscript, however discussion of them to the extent that would require so many figures is lacking, and it is recommended that figures be revisited to only include those that support key points/finds, and follow-up by elaborating on those points. Subsections (particularly in the Results section) disrupt flow, and much of the discussion is included in the Results section rather than in the Discussion section.

This study has great potential, however that potential is left to the reader's imagination. Focusing on and elaborating on the important points (mixing, rivers, significance of C:N, odd 2011/12 year) would greatly improve this manuscript and warrant its publication. The manuscript is written as a simple descriptive paper of the distribution of measurements made—as the title suggests—but much more could be gained if reported data, available (satellite) data, and previous studied were considered and compared more critically.

**General comments**

Paragraphs are often times short and disorganized. In general, paragraphs should be ≥3 sentences long, and flow from one to the next.

Punctuation throughout the manuscript could be improved. Specifically, there are many sentences that are either very long or very wordy that would benefit from including a coma or two.

Many times "well mixed" (and "carbon rich" and "near shore") is used as an adjective, and when it is used that way it should be hyphenated (i.e., should be written as "well-mixed").

When writing numbers, it is good to be consistent. For example, the authors switch between "2 years" and "two years" many times. Generally, it is good practice to spell out the numbers. Exceptions could be dates, concentrations/units, and when doing math.

Include figure citation at the end of the sentence so flow is not disrupted.

It is not good practice to begin sentences with abbreviations and should be avoided.

Many abbreviations (BML, CEND, CRM, CV, DOC, DON, ICES, LOD, POC, PON, SML, SRM, T-S, TDN, TOC) are either not defined, defined after they are used in the manuscript, or their meaning is unclear.

When reporting averages, it seems that medians are randomly used without justification. Either justify why medians are used in those cases, or be consistent and always report mean values.

Often times tenses are incorrect (e.g., when referring to a cruise that took place in 2012, describing what happened on it should be written in the past tense—not present tense). Not as noticeable, but sometimes words are singular/plural when they should be opposite.

 "C:N" denotes C-to-N ratio, so writing "C:N ratio" is technically redundant.

Virtually all sections are divided into subsections, which I feel disrupts the flow of the manuscript and delivery of its message overall, particularly in the Results section. I suggest restructuring manuscript without subsections, reorganizing based on topics mentioned, and add subsections if necessary. Overall, I don't feel there is enough material to warrant subsections, as much of it as currently organized seems to be redundant.

There are many instances where names are written inconsistently. "North Sea" should always be capitalized, while when describing its regions that are referred to (i.e., Denmark, East Anglian coast, East Anglian plume, German Bight, Humber estuary, northern, southern, Southern Bight, Thames estuary, Wash, western, Western Approaches), they do not need to be capitalized. Do not abbreviate "North" as "N." (or "N") since "N" is used to denote nitrogen. When referring to the "North Atlantic" it may be simply refer to it as the "Atlantic" to avoid overuse of the word "North" (i.e., it is understood that the North Sea does not exchange with the South Atlantic).

I do not like that hypotheses are included, as some of them are proven wrong. Perhaps this is not uncommon in journal publications, but I have only noticed this style of writing in proposals. Since this is not a proposal, I suggest restructuring/rephrasing the inclusion of hypotheses, as they may be misleading to readers. Explain how your sampling efforts attempted to address the DOM vs. salinity relationship, seasonal variations, DOM stoichiometry, and anthropogenic/river influences, etc.

**Keywords**

Should you include "North Sea"? Carbon, Nitrogen, and Mixing are broad (i.e., not *key*words).

**Tables**

Inclusion of a table that lists cruise numbers and dates is recommended, which can be referred to throughout the manuscript to improve clarity.

Explain what "SML" and "BML" mean, either in captions or in text. Label based on cruise number, in addition to season/region.

**Figures**

For the most part, I don't feel that the supplemental figures are useful and can be omitted. Moreover, I feel that the number of figures included is excessive considering the limited

number of times they are referred to. A better approach would be to limit to the most important figures and refer to them frequently/when relevant, and omit the others.

Figures are often times referred to as a/b/c/d etc., but figures are not labeled a/b/c/d etc. Please label them.

Figure 1: Referring to a map that shows the North Sea as a system—not exclusively as a sampling grid—would be useful, including labeled geographical regions that are referenced throughout the manuscript (e.g., "northern North Sea" and "German Bight" etc. should be labeled). A bathymetric feature (a break?) can be seen at ~54°N in the west and ~57°N in the east—is this the northern/southern boundary that is referred to in the text?

Figure S1: Unnecessary and can be omitted.

Figure S2: Utilizing the colors better, this figure could be condensed into one panel, included in the manuscript and incorporated into discussion.

Figures 2 and 6: I find these figures to be challenging to interpret and not all that useful. Furthermore, I find that claims made by authors based on these figures are often times incorrect due to the boxes overlapping. I suggest omitting them and interpreting values with respect to depth (and perhaps adding shading to clarify ranges), similar to Figure 7.

Figure 3: Great, clear plots. I suggest doing this with T, S, nutrients, and oxygen. Also be good to include bottom.

Figure 4: Great, but should be reevaluated with regressions (or similar statistical analysis), and a color scheme with more contrast (e.g., red, green, orange, blue) would improve it. Label by cruise number in addition to season/year, and restate what is meant by northern/southern in the caption.

Figure 5: Very nice, and I hope it is a focus of the/a discussion (sub)section.

Figure 6: I suggest omitting and refer to Figure 7 instead.

Figure 7: Would be good to include additional colors to partition by region (northern/southern), and perhaps draw a line to specify surface/bottom samples or use different shapes (diamond = surface, square = bottom).

Figure S3: Why is this figure supplemental? It seems fundament to the discussion of riverine input. Since the lowest DOC concentrations are ~50 µM (not 0 µM), it would make sense for the color to reflect that. Also, please include units for DOC. Salinity also doesn't appear to go below 33.5, or if it does those values can't be seen because they are the same color as the depth colors. Please make these adjustments and include figure in manuscript. Caption should state that they are surface values. Can this be included/interpreted with respect to temperature and nutrients, in a similar manor as Figure 3, and included in the manuscript?

Figure S4: Unnecessary.

Figure S5: There is nothing significant about any relationship plotted, and therefore there is no significant inverse relationship with salinity. This figure is unnecessary.

Figure 8/9: Unnecessary. The same information can be gained by referring to Figure 5.

**Abstract**

DOC and DON are not defined, while dissolved organic matter (DOM) is.

Lines 15-6: "…with higher DOC and lower DON in 2011 and lower C:N ratio and more moderate concentrations of DOC and DON in 2012." is confusing… higher [DOC] and lower [DON] in 2011 = higher C:N in 2011 than 2012, so why not just write that? Added detail on *concentrations* is superfluous unless sentence is restructured/point clarified.

Line 16: Is it necessary to include "differences" twice in the sentence that begins with "Using other data we…"?

**Introduction**

Overall, the introduction brings up some interesting points but does not fully explore them and the papers citing, including them only in lists rather than by understanding and explaining prior, relevant findings.

**Page 1**, lines 24-5: "…have been proposed as potentially disproportionately important for the drawdown of atmospheric carbon to the deep ocean." is very hard to read. "…may be more important for the drawdown of atmospheric carbon to the deep ocean, relative to the open ocean." Is that better?

Line 26: "shelf carbon pump processes" would be clearer as "carbon pump processes on the shelf"

Lines 27-8: Seems redundant to include "complexities" and "complex" when describing the same system. I suggest replacing "complex" with "dynamic" or simply omit.

Lines 23-9: Are these two sentences (hardly a paragraph) necessary to begin the Introduction section? I suggest omitting them and directly begin with a subsection, or elaborate on the points made so paragraph is ≥3 sentences.

**DOM and the continental shelf**

**Page 2**, lines 3-6: By definition, it is not "Marine DOM" if it includes "both terrestrial and marine [material]". Sentence could be rephrased to begin as "In the marine environment, DOM…" or something similar so point on mixture can stay. Is "lifetime(s)" the correct word? Seems that "residence time(s)" is more appropriate. What is "Its" in reference to? Marine DOM? If so, DOM degrades to inorganic carbon explicitly? Perhaps "…its lifetime in relation to degradation to inorganic carbon…" should be written as "…DOC's residence time prior to degradation to inorganic carbon…" Is "(e.g. Jiao et al., 2014)" necessary prior to the end of the sentence? Perhaps due to this citation disrupting sentence flow or a word or two being missing from it, I do not understand what is meant following the citation.

Line 13: Not good to begin a paragraph with "This..." Could you specify what "this" is? Do you mean "The export flux of DOM…"?

Line 14: The "production" itself is exported? Do you mean that newly produced DOM is exported?

Lines 14-5: The portion of the sentence following the semicolon can be omitted, unless examples of previous studies that demonstrated this uncertainty can be provided.

Lines 15-6: Please provide examples of how the stoichiometry may be an important indicator. Including such examples would certainly be good justification for conducting this study.

Line 19: The Redfield ratio is a indeed a "single fixed ratio," defined as 106:16:1 for C:N:P. These ratios have been further evaluated and C:N:P is not always 106:16:1—considering C:N:P does not make it a the Redfield ratio.

Line 20: What is meant by "…as shelf seas process internally-produced organic matter…"? Do you mean "…as autochthonous organic matter degrades/is mineralized in shelf seas…"?

Line 22: Surely dissolved organic *carbon* is carbon rich… Do you mean "Carbon-rich DOM…"?

Line 26: Explain the jargon "refractory" better, in the context of DOM. Perhaps also useful to explain "labile" and why that would conversely not result in the marine environment being a sink for atmospheric carbon.

**The North Sea system**

Could subsection simply be "The North Sea"?

**Page 3**, line 5: 25-30 m C/m^2/yr is highly productive? Provide comparison(s) with other, perhaps better studied, (un)productive systems so reader can grasp relative productivity.

Line 7: Is it "thought" to be or is it "understood" to be net autotrophic? Can you provide an additional citation to better show that efforts have been put forth to understand the system?

Line 8: Can you describe the seasonal stratification better, and how that results in *net* autotrophy? Seems out of place, perhaps due to "(i.e. …)" Perhaps "driven by" would be better than "through its"

Lines 10-1: Can you provide more context on this CO2 flux? What other region(s)/fluxes does this compare to that biogeoscientists/oceanographers might be familiar with? Is "Deeper waters" relative to the overlying waters? The southern North Sea? Please clarify what is meant by "deeper." Do you mean bottom waters, as I suspect they most readily overflow/exchange with the deep ocean.

Lines 12-3: What is "*net* DIC exchanges"? Either provide direction (net to ocean or net to North Sea) or omit "net". Could DIC be introduced earlier? Inorganic carbon is referred to frequently, it seems, so an earlier introduction to DIC may be useful.

Lines 14-7: Elaborating on these "Recent studies" and the interannual circulation variations would be useful for this paper/study. Please do so in a stand-alone paragraph, as this long sentence does not provide reader with enough information.

Lines 19-20: Does the observed "minor net respiration of DOC" contradict the net autotrophy found in the northern North Sea, as stated in the previous paragraph?

Line 22: This decoupling is not "apparent" to reader—please elaborate.

Line 23: Is this "strong seasonality" limited to the northern North Sea? You've previously partitioned the system into northern and southern regions, so any further discussion of the system should specify whether the entire system or its northern/southern regions are being considered.

Line 24: "or so" is vague → omit. Since "weeks" is plural, it could include a month (4 weeks), so stating "weeks to a month" is overly descriptive. Perhaps writing "weeks to months" or "weeks to a couple months" would be the most appropriate wording. An alternative could also be "1-6 weeks".

Lines 26-7: "impossible" is a strong word. Does this sentence suggest that it is impossible to determine whether or not the system is net heterotrophic or net autotrophic? Previous paragraph cited papers that showed that it is/isn't depending on region. Could this point be clarified, perhaps elaborating on distinctions between DOC, POC and DIC fluxes? A better description of the biological carbon pump, in the context of this study/system, would be useful and strongly encouraged.

**Study area, sampling and analytical methods**

Could simply be "Material and methods"

I am surprised that oxygen was either not measured or is not reported here. Why is that?

**Study sites and field sampling processes**

**Page 4**, line 10: Ship names should be italicized.

Lines 10-1: Rather than listing cruises and dates in parentheses, include a table. What does "CEND" mean? An abbreviation for RV *Cefas Endeavour*? If so, write "cruise no." in parentheses so reader is aware that "CEND 14/11" etc. are cruise numbers.

Lines 11-2: Which "two summer" cruises? You *just* introduced the cruises by number and date—refer to the cruises by name/abbreviation so stating that they "The two summer cruises were the summer surveys" is not redundant. "ICES" should be in parentheses, following "International Council for the Exploration of the Sea" (outside of parentheses) and omit redundant "international."

Lines 12-3: "survey…" is used three times in this sentence. Can it be reworded/structured so "survey…" is only included once? What "sampling grid" are you referring to? The one illustrated in Figure 1? What is meant by "survey rectangles"?

Line 16: Can you be more specific than "more northerly stations"? Is there a line of latitude or a bathymetric feature that was not crossed? Which cruise numbers? "2012" and "winter cruise" are vague considering your previous cruise number descriptions.

Line 19: Only "Surface and bottom waters were sampled…"? What about intermediate depths? Figure 7 shows that many intermediate depths were sampled. Or are these sampled intermediate depths simply a result of a shallower water column? Please clarify.

Lines 21-2: If standard seawater has a salinity of 35 and rivers have a salinity of 0, a *minimum* salinity of ~31 and majority of samples with salinity >34 does NOT suggest a "strong riverine influence." It suggests dilution, likely due to riverine influence (and precipitation > evaporation).

Line 25: Was nitrite measured, or was it only nitrate + nitrite? If not, please provide an example of a previous study that showed that nitrite values are negligible for purposes of this study (presumably they are), justifying nitrate + nitrite henceforth being referred to as nitrate. If nutrients are written out (e.g., "ammonium, phosphate …"), "NO3- and NO2-" should also be written out. POC (and PON) should be introduced when describing the biological carbon pump in the introduction section.

Lines 26-8: Surely Tom Hull can provide you with a description of the "standard techniques" so readers are left informed, rather than clueless.

**Analytical procedures**

**Page 5**, line 2: "glass fiber filters, 47 mm diameter of nominal pore size 0.7 µm…" could be reworded as "47-mm diameter glass fiber filters of nominal pore size 0.7 µm (GF/F)…" (Note that the main reason for rewording is to include "GF/F")

Lines 4-5: "This storage regime…" has either not been described in the text or is unclear. Please elaborate on how samples were stored prior to analysis. "has previously been shown to be effectively preserve and not contaminate these analytes" should be rewritten as "effectively preserves these analytes without contamination" (by the way, is "effectively" necessary?)

Line 6: Omit "the water volume recorded." and/or move to appropriate place. The "a" in "Cholophyll *a*" is italicized while it is not on pg 4, line 25 — be consistent.

Line 7: What is meant by "collected from a separate water sub-sample on the same type of GF/F…"? That chorophyll was also collected (from the same Niskin bottle) on a (separate) GF/F? Not clear as written. "fibre" is used while on pg 5, line 2 "fiber" is used. The majority of this seems to be written in British English, so "fibre" should be used in both instances. Please be consistent.

Lines 8-9: "immediately" is vague and likely not true, given its definition of "at once; instantly." I suggest rephrasing sentence to be similar to "All samples were filtered at sea and frozen (-20°C for … -60°C for …) after filtration, until further analysis on laboratory on land." What is meant by "samples"? Are these seawater (liquid) or filters (particulate)?

Line 14: Elaborate on the mysterious "minor developments."

Line 15: "The combustion…" sentence is too short. How can this be included in another sentence/expanded upon?

Lines 18-9: Try to avoid using parentheses whenever possible. "…acidification (adding … 180 s)" could be changed to "…100 µl of 10% HCl was added to 6 ml of sample, spargingwith pure air for 240 seconds and stirring for 180 seconds…" Are the details on time necessary if these are automatic (and presumably default) settings?

Line 33-4: Sentence structure is odd. Should be "… (CRM) were used to verify DOC and TDN measurements: low carbon water…"

**Page 6**, line 3: "Consensus values of DOC for DSR vary in each batch." Seems obvious and unnecessary.

Line 4: "agreement" might be a more suitable word that "accord".

Line 15: "…analysis, the analysis…" → redundant.

Lines 16-7: "in good agreement" loses meaning when used to describe the *exact* same value *and* a value within a range.

Line 31: Since TDN includes inorganic nutrients, the paragraph on dissolved inorganic nutrients should come before the TDN paragraph.

**Pages 6-7**, lines 34-1: "CRM" was previously defined as *consensus* reference materials (page 5, line 33), while Environment Canada provides *certified* reference materials.

**Page 7**, lines 1-2: "filters … desiccator…" should be reorganized as "filters were placed in a desiccator overnight (12 hours) that was…"

Line 7: The detection limit of what? POC or PON or chorophyll or ???

**Results**

Much of this section is discussion and should be moved to the discussion section.

Would be improved if subsections were omitted.

**Physical oceanographic conditions**

Line 10: List cruises by number and refer to suggested table (see previous comment).

Line 11: "biogeochemical" technically includes physical.

Line 15: Why is "Winter" capitalized?

Line 18/Figure S2: If T-S diagrams provide key information for interpreting your data, these plots should not be supplemental. I think they are great and should be included.

Line 19: warmer/fresher/colder/saltier relative to what? Are the "ers" necessary? Makes sense for winter data.

Lines 20-4: "Although…" could be moved to the discussion section.

**Inorganic nutrients**

Lines 26-9/Table 1: Numbers in a table to not "show" what a figure can. I suggest showing these distributions as a figure with (profile or surface map) subplots, or perhaps just refer to Figure 2.

**Pages 7-8**, lines 30-1/Figure 2: I see no reason for silicate to be excluded from Figure 2. Either include or don't bother mentioning. Why aren't data partitioned by region?

**Page 8**, line 4: Where is N:P shown? This should be included in Figure 2 if it is a result.

Lines 5-7: This is discussion.

Lines 10-1: Significant? Please demonstrate numerically/statistically.

Lines 14-8, 21-5: This is discussion.

**DOC and nitrogen concentrations**

**Pages 8-9**, lines 31-8: This is, for the most part, discussion.

**Page 9**, line 1: I disagree that hypothesis 1 is "confirmed" based on a salinity gradient of 31-6. I suggest reevaluating DOM-salinity relationships in the context of mixing/dilution rather than rivers. Perhaps if a riverine end-member is used this can be assessed, but "confirm" is a very strong word. Ducklow et al. (2007) and Margolin et al. (2016) used a riverine end-member approach for the Black Sea that may be useful to consider, if an end-member is available. Ducklow et al. (2007) also considered C:N stoichiometry.

Line 13: DOC is virtually always at least one order of magnitude lower than DIC – this is obvious and cited example is not needed.

Line 14: Actually, they are *approximately* 6 (approximately six or ~6) times smaller.

Line 15: How is this further demonstrated? Is there an example/citation to compare to? This is getting into discussion territory…

Lines 20-9: For the most part, this would fit better in discussion.

**DOM stoichiometry**

**Page 10**, line 3: Where does the "expected North Atlantic endmember of 13-15" come from? Please cite and explain how that is expected.

Lines 5-7: Omit hypotheses as this is not a proposal. Furthermore, this is not results—it is discussion! Section 4 is the discussion, so presumably most things mentioned in the results are discussed there!

Line 9: A gradient of 6.5-7 is like a C:N of 106:15-16, which is very similar to, if not the same as, Redfield (106:16). Using those numbers (i.e., 106 for C) makes the comparison to Redfield easier and readers will more readily grasp that.

Lines 9-11: This sentence is discussion.

Line 8: What is "low" salinity? Relative to other samples? Virtually all samples have salinities > 31, which is high compared to many seas.

Lines 11-2: DOC in surface waters of the open ocean can reach ~70 µM, and the lowest salinities in Figure 4 have DOC concentrations of ~60-120 µM, which is *not* "more" than an order of magnitude. The sentence included here is false and misleading to readers.

Line 15: Again, 106:16 is much clearer, as well as more precise.

Lines 14-20: This is discussion.

**Interannual differences**

Lines 23-8/Table 2/Figure 6: I do not see "northern stratified" "northern surface" or "bottom waters" anywhere in Table 2 or Figure 6, making this text impossible to understand/interpret in this context.

Line 30: I agree with this sentence, with exception of Jan 2012, despite not understanding what "SML" and "BML" are. However, the following sentence states it *specifically* rather than generally. I'd omit the "Generally" sentence because it makes the following (better/more descriptive) sentence redundant.

**Page 11**, lines 1-9: This is, for the most part, discussion.

Lines 14-7: If your hypothesis is "unfounded" it should not be included in a publication. Explain to readers what is gained from the data—reporting what is unfounded demonstrates incomplete interpretation of data. Furthermore, this discussion does not belong in the results section.

**DOM in bottom waters**

Line 20: The word "interesting" does not belong in the results section—it is an opinion and belongs in the discussion section.

**Discussion**

**Pages 11-2**, lines 30-4: A better place for these sentences may be the introduction, if they were to be rewritten slightly. At this point in the paper, the reader should already be aware of these points. Perhaps the abstract and/or conclusions would be a better place than introduction.

**DOM-Salinity relationship**

**Page 12**, line 6: A salinity of ~30 is not "low salinity estuarine waters" This is simply a sea.

Lines 7-8: Again, DOC concentrations in the North Sea are *not* an order of magnitude higher than surface values found in the North Atlantic (~70 µM), but they are *slightly* higher (or perhaps double). A more convincing point is the gradient in surface DOC and DON concentrations shown in Figure 3, which clearly show that the waters further on the shelf/in shallower waters are more enriched in DOM.

Lines 12-4: Where does this information on the North Atlantic Bight come from? There should be a citation with this information. Can this NAB relationship be tested in the North Sea? That seems like an interesting discussion point.

Line 16: Can these "other controls on the concentrations of DOM" be elaborated on/discussed further? This is, after all, the discussion section!

Line 18: What is hypothesis 2 again? I think it would be better to explicitly state the point rather than refer to hypotheses.

**Significance of difference in DOC inventory between 2011 and 2012**

Line 26: How is it significant? Statistically *and* biogeochemically speaking.

Lines 27-8: By "apparent" do you mean "average"? If average is what is meant, please be specific regarding average since both mean and median have been used throughout the manuscript. Was this change an increase or a decrease? Looking at Figure 6, DOC boxes overlap slightly in the stratified plot, while the error lines do in the summer mixed, so I'm not sure how "significant" these differences are. By also considering Figure 3, it is clear that the concentrations decreased for both DOC and DON, so this figure should be referred to.

Line 31: What percentage decrease is 10-20 Tg relative to the North Sea's DOC inventory? That seems like a useful and interesting way to interpret these numbers.

Line 32: This comparison with DIC is interesting, although you are referring to a decrease over a year (Aug-Aug) while it seems Thomas et al. are referring to what is presumably an increase in 30 Tg C/year that is then consumed/replenished (i.e., in balance). Why is your change so large? Was it just an odd year? These points should be elaborated on!

**Page 13**, line 2: If "our" best estimates, why do you cite Thomas et al? Please explain where the "our" (your interpretation of Thomas et al's data?) comes in, and where Thomas et al. come in.

Lines 3-4: Yes, just for the years concerned… This is potentially very interesting! Why is this paragraph so short? It seems there is much to be discussed here regarding the contrasting Augusts. Are there climatological effects that would result in this, such as El Niño or NAO? Consider exploring satellite chlorophyll data before/after 2011/2012 to see if one of these years is anomalous or if there is a trend. There is much to explore and discuss, but where is that in this discussion section?

Lines 9-10: High DOC coinciding with high salinity does not suggest that rivers are important, contradicting previous claims, as far as I understand.

Lines 13-5: This sentence is very hard to understand since the previous sentence referred to low DOC, and this sentence begins with high DOC. It is unclear whether authors are suggesting that high DOC values are coming in from the ocean or leaving to the ocean (or something else?)

Lines 15-6: "observations…observed" → redundant. Is there a figure or table that could be referred to in order to guide the reader?

Lines 13-24: These sentences are a really great part of the discussion, although I feel much more thought could be put into the points made, as this is where new insights and understanding comes from. It is disappointing to see this subsection end/a new one begin just as new understanding begins to happen. "confused" has previously been used to describe a graph, and I'm not exactly sure what it means. Do you mean unclear? How are riverine inputs important? Looking at Figure S3, and am not convinced that they are. Could the distribution in Figure S3 be due to it being in the winter? What does this distribution look like in the August cruises? These points need to be discussed further before changing to a new subsection of the discussion.

**DOM variability, C:N ratio and the seasonal signal**

Line 27/Figure 5/Table 2: What are the $R^2$ values for the correlations? I agree that the correlation looks fairly good for the 2011 data, but it does not look great for 2012. More details are needed! Would be better to include regression equations (and $R^2$ values) in text and/or in a table. C:N is as low as 5.9 and high as 36.5 in Table 2, so "roughly 9 to 17" is incorrect.

Line 30: Overall this paragraph is too short, and this sentence lacks discussion of comparisons made in previous sentence. What is the significance of comparing C:N between these systems? This needs to be discussed further here.

Lines 32-3: This sentence is unclear, and perhaps does not make sense. I don't see how Figure S5 supports what it stated here, as there is nothing significant about any relationship in Figure S5, as $R^2 = 0.15$ means there is no significant relationship, and therefore no inverse relationship.

**Page 14**, lines 7-12: This could be discussed more.

Line 15: Why does it matter that you predicted this? It is not surprising that surface DOC is higher than deep concentrations, especially in the summer.

Lines 17-8/Table 2: Where are surface and/or summer values listed in Table 2? This is unclear, and renders table useless, especially if values are given in text. Either omit values in text and clarify table, or omit table (I suggest the former).

Lines 19-20: If this citation list is going to be included, the reported findings from listed papers should be compared/contrasted with yours, as this list means nothing to a reader, other than that if they want information they should look elsewhere.

Lline 21: Surely it is not impossible. This sentence is very wordy, which does not serve the manuscript well.

Lines 24-5/Figure 6: I disagree, as I don't see any portion of the figure labeled as "northern" and many of the bars overlap, so it doesn't seem there is much of a difference.

Lines 24-8: I feel like this either has been covered, or should have been covered, in the results section—where is the deep, exploration and interpretation of your results?

Line 28: Where is Figure 9?

**Pages 14-5**, lines 33-4: This sentence is too long. Discuss what this means/what can be learned from it.

**Page 15**, lines 7-12: This sentence is too long, and I don't see the point in mentioning all of these processes since results from other papers are not considered, and discussion/deduction of what the likely scenario is is lackling — rather you only provide this list of possibilities. These points need to be expanded upon and explored as a part of the discussion.

**Conclusion**

Lines 31-2: I'm sorry, how does this support the claims from Barrón and Duarte? What were their claims again? DOM negatively correlates with salinity is their major finding? I don't see the significance of this, as this trend is common.

**Page 16**, lines 2-3: What freshwater end-members? Freshwater means salinity = 0!

Line 13: Should be "high C:N DO*M*"

**References**

Ducklow, H.W., Hansell, D.A., Morgan, J.A. 2007. Dissolved organic carbon and nitrogen in the western Black Sea. Marine Chemistry 105: 140-150, https://doi.org/10.1016/j.marchem.2007.01.015.

Margolin, A.R., Gerringa, L.J.A., Hansell, D.A., Rijkenberg, M.J.A. 2016. Net removal of dissolved organic carbon in the anoxic waters of the Black Sea. Marine Chemistry 183: 13-24, https://doi.org/10.1016/j.marchem.2016.05.003.

---

## Author Comment (AC1) · 6 Sep 2018

We thank both reviewers for their in-depth reviews of our manuscript. Given the overlapping nature of the various comments, and the resulting re-write of the manuscript, we provide responses to both reviews in the attached zip file, along with a revised manuscript and revised supplementary material. We provide both a clean and a 'track changes' version of the revised manuscript to aid the reviewers and editor in assessing the changes we've made. We apologies for the long delay in our final response.

Please also note the supplement to this comment:

https://www.biogeosciences-discuss.net/bg-2017-387/bg-2017-387-AC1-supplement.zip

---

## Author Response (AR2)

**Chaichana et al: Response to reviewers**

We thank the reviewers for their comprehensive reviews and feel that the resulting manuscript is substantially improved. We reposed to their comments below. Reviewer's comments are in italics, our responses in red. Overall we

5 have reworked the manuscript to remove hypotheses, introduce a new synthesis of literature end-member concentrations to aid the analysis of DOM concentration changes over salinity gradients and restructured and extended the discussion, in line with the reviewers suggestions. A 'track changes' version of the revised manuscript can be found below the response to reviewers.

**10 Reviewer 1**

**General Comments**

This paper details a multi-year high-resolution sampling of DOM dynamics in the North Sea. The main conclusions of the work suggest that there is high spatial and temporal variability in the total concentrations and C:N ratio of OM over the sampled periods, and that this inter-annual variability has

- 15 strong implications for overall carbon budgets in the region. The implications of this work for carbon cycling in the region are important, and I found the paper to be generally well written. However, I struggle with the chosen focus on C:N ratios for elucidating DOM dynamics. I felt that novel portion of this work, that is, the elucidation of the impact temporal variability has on the overall carbon budget(s), was not emphasized enough.
- 20 We thank the reviewer for the positive summary of the paper and are pleased that they recognize the importance of this dataset, which we believe is of value and interest to the community. We agree with the reviewer that the inventory changes should have greater prominence in the manuscript and have tried to address this, as detailed in the responses to specific points below and in the manuscript. However, we would defend our approach of using C:N stoichiometry as a key indicator of the biogeochemical status of
- 25 shelf waters given the relatively contsant C:N ratio of the open ocean (out of the surface layer) and the low C:N of river inputs to the North Sea, the high C:N ratios observed, particularly in the first summer are, we argue, an important indicator of the state of the system. We address this more specifically in responses below and in changes to the text outlined below.

In particular, there are a few main facets of the work I feel need further development if they are to be

included in the final manuscript:

1) The objective that the C:N ratio is presented to address- and how this is interpreted and discussed. As mentioned in the work, many factors can affect this ratio! If this is the only metric you use to assess DOM dynamics, you need to be really careful. Can you really answer the larger objectives you outline with the

5 hypotheses that you pose? How does the C:N ratio compare to the chla concentrations, nutrients, and previous work in the region on tracking allochthonous vs autochthonous sources of OM? How does your end-member calculations compare to actual end-members from the literature, and OM composition from other work?

We absolutely agree that the CN ratio is highly variable (and state this in the manuscript), as a

- 10 consequence of the variable nature of DOC and DON. This is why we feel it is a useful indicator of the state of the system. We do not attempt to make any concrete inferences about the source / sink of DOC vs DON in the paper, because we do not have the appropriate observations at our disposal to do so. We had already looked at relationships with chlorophyll, nutrients etc but there is no evidence of significant relationships across the whole data set or in sub-sections of it. Therefore we did not present this data in
- 15 the paper.

Regarding the tracking of different sources of DOM, this was not really our initial aim, but have now included more discussion of this throughout the paper and have also conducted a synthesis of endmembers for the region (both river and open ocean) and furthermore compare these, and our data, with the global synthesis effort of Barron and Duarte (2013) and other temperate shelf environments. This has

20 provided a new and useful conceptual framework for the paper which hopefully addresses both reviewers' concerns and we feel has added usefully to the paper. In particular see changes to Sections 1.2 and 4.6, Tables S1 and S2 and new Figures 1 and 9.

2) Tie the POM work in better. How does this compare to the DOM, and what is this impact on the overall carbon budget?

25 It would be really good to do more with the POM but unfortunately we only have POM data for the second year of the study and therefore cannot compare the differences between the 2 years. We have included the POM data primarily to demonstrate that it is a small component of the budget, which is in line with other studies. In particular, the 'missing' carbon in 2012 cannot be stored in a large stock of POC – which

we can see is not present. We have stated this explicitly in the final paragraph of section 4.4.

The discussion and conclusions regarding the relationship of DOM with salinity. This relationship has been found to be conservative with mixing of the major water masses in the North-Baltic Seas transition, however major non-conservative processing on DOM does occur in the region (see eg. Osburn and

5 Stedmon, 2011 Marine Chemistry DOI: 10.1016/j.marchem.2011.06.007). I would like to see further development of this, including tying the current work into previous analyses of DOM composition, etc. What are the implications of local variability, in this context?

We have included much more discussion of DOM-salinity relationships in the revised manuscript as we agree with the reviewer that it is important. However, we would point out that the North Sea is under a

- 10 very different regime to the Baltic. The Baltic is an enclosed sea, with circulation dominated by salinity and a limited exchange of water with the ocean (the North Sea in this case). The North Sea however, is dominated by 'through-flow' of Atlantic water, via the English Channel and Malin shelf and out through the Norwegian trench. As such the open North Sea is subject to much smaller ranges of salinity (typically 30-35) and autochthonous processes are likely to be much more dominant. We make this case in the
- 15 introduction for the sake of clarity we would not expect strongly conservative behaviour and we do not believe that a significant component of the DOM in the N. Sea is of terrestrial origin in the summer, both because of the minor influence of freshwater on the main part of the North Sea but also because the long residence time will favour loss of terrestrial organic matter before reaching the higher salinity regions. This is discussed in Sections 1.2, 4.1 of the new manuscript.
- 3) This is perhaps the most important- I feel the discussion on C inventory should take center stage.We agree, and have made specific reference to this issue in the title of the piece and have expanded this section in the discussion.

How does this work advocate for or against high-resolution measurements?

25

We do not feel that this finding particularly advocates for or against high resolution measurements (although clearly they are often desirable) so have not commented on this.

How does it revise or promote our understanding of DOC cycling in the region? This is now covered in the discussion and conclusions section. In brief, DOC is probably the most variable pool of carbon in shelf water columns, given the inventory change observed here. The North Sea does not appear to be net-heterotrophic (certainly not in all years) as previously suggested by Bozec et al. The variable flushing time of the North Sea seems likely to be the control on DOC concentrations and stoichiometry.

**5 How does this compare to other regions and/or the global budget?**

We have introduced comparison with the global synthesis of Barron and Duarte (2015) and Vlahos et al study of the Mid-Atlantic Bight (MAB) section 4.2, Fig 9. This highlights 1) that the DOC concentrations in the North Sea and MAB are similar to each other and comparatively low at a given salinity compared to the global shelf average and 2) that the inventory change between the two years is therefore all the

**10 more significant for the role of DOM in the carbon cycle in shelf seas.**

What are the uncertainties with C budgets, and does this paper help to narrow these? We don't feel that this paper is the place to assess the uncertainties in carbon budgets of the North Sea however we do put our results in the context of the Thomas budget in sections 4.3, 4.5 and 5. Our study (as others also have) suggests net autotrophy and a DOC source to the open ocean, in contradiction to the

15 Thomas et al 2005 budget; so we are cautious about discussing 'narrowing' of uncertainty. The important point that we do make is that the dynamics of DOM clearly have a very important role to play in the export of carbon from the shelf.

**Specific Comments**

**20 1 Introduction**

The first paragraph starting on line 23, to me, is the motivation for this work.

We agree and are pleased to keep this paragraph despite reviewer 2's recommendation to remove it.

Clearly relate the following discussions, and set-up to the goals of the study relative to this. Be explicit upfront- what do you hope to find with this work, and how is it novel (e.g. lines 25-29)? Some of this is

25 outlined in later the introduction, but the narrative back to the main objectives of the work is lost throughout the rest of the paper.

We have now clearly stated the aim after the opening paragraph of the introduction, and also made it clear the constraints of the study – that it results from a PhD study on cruises of opportunity: 'In this study we

investigate the variability of the organic carbon inventory of a large and complex shelf sea by considering the evidence provided by two high-spatial-resolution summer surveys of organic matter concentrations, stoichiometry and deviation from conservative mixing between river and open ocean end members. These data were collected during cruises of opportunity during a PhD research programme, so our analysis is focussed on diagnostic,

5 geochemical approaches to understanding bulk concentrations and intentionally does not attempt to elucidate distinct sources or types of DOM, or determine process rates, given only *prima facie* evidence. 'We have not stated what we 'hope to find' as this seems pre-emptive of the findings.

we have not stated what we hope to find as this seems pre emptive of the midnings.

The authors attempt this tie by stating hypotheses and referring to them throughout. This causes the prose

- to be a bit awkward- and I suggest instead outlining the overlying research outcomes the authors hope the study will answer. As is, the hypotheses are too narrow to the stoichiometry work, and don't really address our gaps in understanding of the temporal variability of the C cycling in the region.
   We have removed the hypotheses and replaced them with aims and predictions in our re-structured introduction. Reference to hypotheses later in the paper are removed.
- 15 Perhaps a figure would help with synthesizing what we know, and what gaps this study addresses? We have considered this carefully, but in the end feel that this would start to make the paper more like a review paper than a report of field data and subsequent analysis, so have not made this change. We do however feel that the new Figure 1, presenting idealised salinity gradients and expected C:N ratios sets up the paper rather better than previously and goes some way to a synthesis figure at the start.
- 20 How has this sort of "high-resolution" work refined carbon budgets in other regions?
   To our knowledge regional surveys like this have not been conducted in other regions so we are unable to comment.

What are the processes affecting atmospheric CO2 draw down in regions such as shelf seas? We found that in order to bring this into the introduction in a meaningful way resulted in another long

25 paragraph that seemed tangential to the main point of the paper and so haven't included it in the end – again, if this were a review paper we would be dealing with such topics in great detail but for the purposes of this paper we feel it would be excessive detail.

How does this relate to the global carbon budget?

5

Again, we feel that going into more detail than we already do would require adding considerable additional length to the paper. Our opening sentence was chosen carefully to provide the context and references necessary for the reader to investigate this themselves if interested: "Coastal and shelf seas are generally more productive than the open ocean (Jickells, 1998; Simpson and Sharples, 2012), and through

5 various processes have been proposed as potentially disproportionately important for the drawdown of atmospheric carbon to the deep ocean (Bauer et al., 2013; Regnier et al., 2013; Thomas et al., 2005; Tsunogai et al., 1999)"

Why is the North Sea an ideal system to study in this regard?

20

The North Sea is complex and interesting but it is far from 'ideal'. We have extended the North Sea

10 description to cover a number of the points that the reviewers have raised, which we respond to below.We feel that this now sets the scene more effectively for the paper.

Page 2, Line 6- Is DIC really the only place for long term carbon storage in the ocean? Better tie in why you are looking at DOC, not DIC.

Actually we were making the opposite point – if it isn't broken down to DIC then it's stored for longer

15 (lifetime of refractory DOC >> DIC in the global ocean). We have tried to clarify: "The reactivity of DOM and in particular its availability for breakdown by marine microbes is a key factor controlling its resistance to degradation to inorganic carbon and thus capacity for long-term carbon storage as refractory, i.e. unreactive, DOC on a timescale of hundreds or thousands of years (e.g. Jiao et al., 2014)."

Page 2, Paragraph starting line 13- I would argue that C:N stoichiometry is a very small part of understanding allochthonous vs autochthonous OM sources, and especially reactivity of DOM.

We entirely concur, and don't believe that this paragraph suggests otherwise. We do not go on to use C:N
values directly as diagnostic of source or DOM reactivity. The point of this paragraph is to outline why
DOM stoichiometry might change in shelf seas, which is relevant as we do see and discuss C:N changes
in the results and discussion sections; and to outline how changing stoichiometry (for whatever reason)

25 may lead to changes in shelf carbon pump efficiency. This paragraph is substantially changed in the general re-working of the paper.

You acknowledge some caveats here, but how does more compound specific work (such as isotopes, biomarkers, etc) compare to C:N ratios (e.g. Kaiser and Benner, 2012 JGR-Oceans DOI:

**10.1029/2011JC007141)?**

The point we were making was about the large-scale recycling of N relative to C and carbon cycle response. In this context we don't feel that the isotope / biomarkers work is particularly relevant and to introduce them would add excessive length to the paper.

- 5 Convince the reader that the stoichiometry is an adequate tool for the objective you are outlining, ie. using C:N ratios to understand DOM source and reactivity.
  As is, I feel that the discussion and implications of the work rely too heavily on this.
  We feel that the reviewer has misunderstood this aspect of our paper at no point do we state that we are trying to understand the source or reactivity of DOM based solely on the C:N ratio. We do speculate on
- 10 the origin of the carbon-rich DOM in deep waters in 2011 in the paper, but do so on the basis of a range of evidence and reasoned argument rather than simply relating C:N to source and reactivity. We should have been clearer, and have added the following paragraph to this part of the introduction in order to deal with some of these issues.

'The carbon to nitrogen ratio of DOM therefore has the potential to be a useful diagnostic of the state of the shelf

- 15 system indicating the efficiency of nitrogen re-use throughout the microbial food web. However it is complicated by the interactions with river inputs at differing concentrations and probably variable stoichiometry, as well as seasonal and interannual variability. Some studies have used other parameters to elucidate sources of DOM, for example biomarkers and isotopic signatures (e.g. Kaiser and Brenner, 2012) or spectroscopic or fluorescent signatures (e.g. Painter et al., 2018). In some systems the N:C ratio can be used to determine the relative
- 20 contributions of terrestrial and marine sources to DOC, if endmembers are known and conservative mixing can be assumed (Perdue and Koprivnjak, 2007). In the absence of such techniques, the use of C:N alone as a diagnostic variable in shelf seas must be approached with caution. We explore in this paper its potential application to the North Sea case, in the context of other oceanographic measurements and observations.'

**25**

**Page 3 line 15- Add "climate" in front of cycles*Done (although this sentence has moved in the restructuring of the introduction)**

**2 Methods**

In general, I feel the methods are well explained and analytically sound. An interesting paper recently came out in EST Letters that I feel the authors could benefit from regarding "DON" calculations-Saunders et al., 2017 DOI: 10.1021/acs.estlett.7b00416

We are aware of this work and does of course play an important role in interpreting the uncertainty in

5 DON measurement, which are already incorporated into our data processing, and we have historically taken the same approach as Saunders et al in our work, prior to the publication of their manuscript – hence not citing it. We now cite their paper in the relevant paragraph.

Page 4, lines 21-22- Why did you exclude the riverine-influenced sites? How does this impact your further discussion of sources and end-members and your hypotheses above?

10 As this study was conducted on a ship of opportunity, we did not have the opportunity to choose sampling locations, otherwise we would have sampled up the estuaries. We have changed 'Sampling focussed on the open waters...' to 'The cruise track focussed on...' to make this clear.

3 Results-

15 The results section includes a bit of interpretation in it (e.g. see paragraph on page 11 lines 11-17) and should be reworked to include only observations of the data.

This discussion relates to a hypothesis so the second half of the paragraph has been removed and the first part, describing the C:N ratios has been moved to join the paragraph preceeding it in the restructuring to avoid short paragraphs in response to reviewer 2.

- 20 Do you have the TS profiles? What about other property/property plots? We do not readily have access to complete T-S profiles for the cruise. We feel they would have been of limited additional value over the Temperature and Salinity data associated with the water sampling depths, unless we were to undertake detailed physical oceanographic analysis, which we felt was out of scope of this paper. We present numerous other property-property plots throughout the paper (i.e. original
- 25 manuscript Figures 4,5,7,8). In the process of the analysis we produced many more (DOC vs nitrate, DON vs Chlorophyll *a* etc) but have only presented those which show the most interesting and meaningful relationships, or absences of such, were of interest.

Page 7 line 21- What do you mean by noisy?

We meant that there is a lot of variability around the apparent mixing line in T-S space. This has been removed in the general reworking of the paper.

Page 8, lines 8-12- This is confusing, but is an important distinction. Be clear with your comparisons

5 here, and throughout the rest of the manuscript! Perhaps delineating the water bodies by type for comparisons of measurements over time (eg. Open Atlantic water)?

We agree that this could be clearer. We feel that we largely have delineated water bodies by type (Southern well-mixed, Northern Bottom, Northern Surface). Note that none of the water sampled is the Open Atlantic... The distinction here is that there is a body of water north of 59N but still in the North

- 10 Sea which was only sampled in 2011 and not 2012. We have reworded as follows: 'We investigated the potential bias due to the more northerly extent of the 2011 cruise potentially sampling waters richer in nitrate and phosphate. However statistical comparison (by t-test, p